# Expression of FoxP2 in the basal ganglia regulates vocal motor sequences in the adult songbird

Lei Xiao [1], Devin P. Merullo [1], Therese M. I. Koch [1], Mou Cao[2], Marissa Co [1], Ashwinikumar Kulkarni[1], Genevieve Konopka [1] & Todd F. Roberts [1]✉

Disruption of the transcription factor FoxP2, which is enriched in the basal ganglia, impairs vocal development in humans and songbirds. The basal ganglia are important for the selection and sequencing of motor actions, but the circuit mechanisms governing accurate sequencing of learned vocalizations are unknown. Here, we show that expression of FoxP2 in the basal ganglia is vital for the fluent initiation and termination of birdsong, as well as the maintenance of song syllable sequencing in adulthood. Knockdown of FoxP2 imbalances dopamine receptor expression across striatal direct-like and indirect-like pathways, suggesting a role of dopaminergic signaling in regulating vocal motor sequencing. Confirming this prediction, we show that phasic dopamine activation, and not inhibition, during singing drives repetition of song syllables, thus also impairing fluent initiation and termination of birdsong. These findings demonstrate discrete circuit origins for the dysfluent repetition of vocal elements in songbirds, with implications for speech disorders.

[1] Department of Neuroscience, UT Southwestern Medical Center, Dallas, TX, USA. [2] Department of Pediatrics, UT Southwestern Medical Center, Dallas, TX, USA. ✉email: Todd.Roberts@utsouthwestern.edu

Basal ganglia circuits play important roles in learning and controlling vocalizations[1–3]. The transcription factor forkhead box protein P2 (human FOXP2/zebra finch FoxP2) is enriched in striatal medium spiny neurons (MSNs)[4,5], and genetic disruptions of FOXP2 cause Childhood Apraxia of Speech, a developmental speech disorder marked by difficulties in controlling and accurately sequencing vocal motor actions[6–8]. FoxP2 has also been found to be important for vocal learning in songbirds, indicating that it may have a common functional role in vocal motor control across vocal learning species[9–12]. Nonetheless, how FOXP2 and basal ganglia circuits contribute to the fluent production of vocalizations is still poorly understood.

FoxP2 is strongly expressed in the songbird Area X, a specialized song nucleus within the striatum that is important for song learning and the control of song variability[13–17]. Its expression is thought to be necessary for learning song during development but not for the maintenance of song in adulthood[10,11,18]. Zebra finch song consists of repeated introductory notes followed immediately by ~3–7 syllables that are produced in a highly stereotyped sequence or syntax, referred to as a song motif[19]. This song motif is typically repeated two or more times in what is referred to as a song bout. Juvenile male zebra finches learn their song by imitating the song of an adult male bird, typically their father, during development. Knockdown of FoxP2 in Area X of juvenile zebra finches causes a variety of vocal learning deficits, including inaccurate syllable imitation, reduced stereotypy of song syntax, and anomalous repetition of song syllables[10,11,20].

FoxP2 appears to have a more restricted role once song is learned. A core feature of zebra finch song is low trial-to-trial variability in the acoustic structure of song syllables[21]. Knockdown of FoxP2 expression in Area X of adult birds has not been reported to cause a deterioration of singing behavior or problems with song syntax[11]. However, FoxP2 expression levels are inversely correlated with trial-to-trial variability of song syllables (low levels of FoxP2 are associated with increased song syllable variability) and FoxP2 has been linked to gene expression modules associated with vocal variability[12,22–24]. In addition, male zebra finches decrease the trial-to-trial variability of their song by ~1–3% when singing to a female bird, a performance mode known as directed song, when compared to undirected singing[19,21], and knockdown of FoxP2 blocks this social context-dependent decrease in song variability[11].

Several lines of evidence indicate that the function of dopaminergic input to Area X and the behavioral role of FoxP2 expression are tightly linked. Like the mammalian striatum, Area X receives glutamatergic input from pallial/cortical regions and input from dopaminergic neurons in the substantia nigra and ventral tegmental area (VTA)[25]. FoxP2 is thought to exert its effects in part by regulating the expression of D1 dopamine receptors in MSNs. Disruptions of FoxP2 expression result in reduced expression of D1 dopamine receptors in Area X[11] and have also been shown to increase dopamine levels in the rodent striatum[26]. Ablating dopaminergic input to Area X severely disrupts song learning in juvenile birds but does not drive deterioration of overall song acoustic structure or syllable syntax in adult birds; these behavioral effects are similar to those associated with disruption of FoxP2 expression in juvenile and adult birds[10,11,27,28]. Moreover, like disruption of FoxP2 expression, 6-hydroxydopamine lesions of dopamine terminals in Area X reduce trial-to-trial variability of song syllables during undirected singing[29], and pharmacological inhibition of D1 receptors blocks social context-dependent decreases in song variability[30].

Together, these previous studies suggest that FoxP2, D1 receptors in MSNs, and dopamine input function in concert to regulate song syllable acoustic structure and vocal variability. We were motivated to further explore the function of striatal FoxP2 and dopamine in adult vocalizations because several additional lines of evidence indicate that we still have an incomplete view of their role in controlling vocal motor sequences. First, Childhood Apraxia of Speech resulting from genetic disruptions to FOXP2 is characterized by difficulty in sequencing the phonemes needed to form words and speech[31]. In addition, heterozygous Foxp2 mutant mice do not exhibit changes in the acoustic structure of their ultrasonic vocalization syllables but do exhibit disruptions in sequencing syllables[32,33]. Second, previous FoxP2 knockdown studies in Area X have relied on viral expression using lentivirus, which only sparsely infects avian neurons[10,11], and have been carried out using non-reversible viral approaches which can limit the ability to fully attribute changes in vocal behaviors to changes in FoxP2 expression. Third, Area X has been implicated in song initiation, syllable sequencing, and song maintenance. Neurotoxic lesions in Area X can induce transient repetitions of song syllables and problems initiating song bouts[34,35]. Deafening-induced deterioration of song is blocked by lesions of Area X, suggesting an active role for Area X in the maintenance of adult song[15]. Lastly, viral expression of the Huntington's disease mutant gene fragment in Area X, which leads to MSN cell death, causes repetition of song syllables and disruptions in the sequencing of song[36].

Here, we apply novel reversible manipulations of gene expression and neural circuit activity to re-examine the role of striatal FoxP2 and dopaminergic circuits in the control of adult vocalizations. In contrast with previous studies, we find that FoxP2 is critical for the fluent initiation and termination of adult song and regulation of song sequencing. Adeno-associated virus (AAV) mediated knockdown of FoxP2 using a small hairpin RNA (shRNA) reliably causes an increase in the repetition of song syllables. Turning off viral expression of the shRNA two months later rescues these disruptions in song. To clarify connections between FoxP2 and dopaminergic circuitry, we mapped all Area X cell types and the distribution of FoxP2 and dopamine receptor expression using large-scale single-nucleus RNA sequencing. We find that FoxP2 knockdown causes a decrease in the expression of the D1 and D5 subtypes of dopamine receptors in Area X direct-like pathway MSNs. These results, combined with previous studies indicating that mice carrying different substitutions in their endogenous Foxp2 gene exhibit specific alterations in dopamine levels in the striatum[26,37], encouraged us to test whether manipulation of dopamine release in Area X disrupted sequencing of song syllables. We find that optogenetic excitation, but not inhibition, of VTA axon terminals in Area X during song production also disrupts the fluent initiation and termination of adult song and that these changes in song are dissociable from reinforcement-based changes in song syllable acoustic structure[38]. Together, these findings indicate that FoxP2 regulated expression of dopamine receptors and changes in dopaminergic signaling in the striatum work in concert to control the fluent production of learned vocalizations.

## Results

**FoxP2 Knockdown disrupts syllable sequencing and repetition.** We used a Cre-switch (CS) AAV platform to drive a broad and genetically reversible knockdown of FoxP2 in adult zebra finch neurons[39] (Fig. 1a). A single transgene containing open reading frames of two fluorophores (mCherry and tagBFP) oriented in opposite directions switches between expressing mCherry or BFP depending on Cre-driven recombination. Short hairpin RNAs against the zebra finch FoxP2 gene (shFoxP2) or a scrambled control hairpin (shScr) were inserted into the 3′UTR of the mCherry in the functional orientation with respect to the CAG promoter[40], allowing expression of the shRNA to be turned off with the introduction of Cre recombinase[39,41].

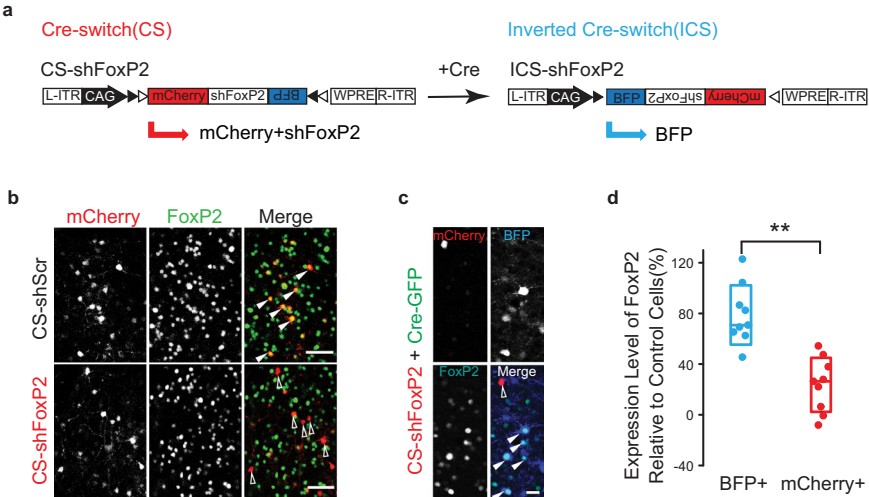

**Fig. 1 A Cre-switch platform for testing the function of FoxP2 in learned vocalizations. a** AAV constructs to achieve Cre-dependent silencing of zebra finch FoxP2. For the Cre-Switch (CS) configuration, the expression of mCherry and shRNAs are only maintained in the absence of Cre, whereas the expression of tagBFP(BFP) is activated in the presence of Cre. WPRE, woodchuck polyresponse element. ITR, inverted terminal repeats. Open and filled triangles indicate loxP and lox2272 sites, respectively. **b** FoxP2 expression in the cells infected with CS-shScr (top, filled triangles) and CS-shFoxP2 (bottom, open triangles) in Area X of adult birds. mCherry+ cells are cells infected with CS constructs. Scale bar, 50 μm. **c** Confocal images showing the expression of FoxP2 in mCherry+ (open) and BFP+ (filled) cells within Area X of an adult bird injected with CS-shFoxP2 and Cre-GFP constructs. Scale bar, 20 μm. **d** Quantification of the expression level of FoxP2 in BFP+ and mCherry+ cells (blue 78 ± 18% vs red 23.7 ± 16.4%[mean ± SEM] relative to control cells) within Area X of adult birds injected with CS-shFoxP2 and Cre-GFP constructs (n = 9 slices from 3 birds). The expression level of FoxP2 in BFP+ cells is significantly higher than in mCherry+ cells (p = 0.0002, Mann–Whitney test). Each point represents the normalized expression level of FoxP2 in BFP+ or mCherry+ cells relative to control cells from individual slices. Box indicates the median ± SD.

We tested CS-shFoxP2 and CS-shScr constructs in vitro and in vivo and found that we could significantly reduce the expression of FoxP2 following expression of CS-shFoxP2 (Fig. 1b and Supplementary Fig. 1a, b). In contrast, the inverted Cre-switch (ICS) configuration of shFoxP2 resulted in no change in the expression level of FoxP2 in vitro (Supplementary Fig. 1c). Further testing of this configuration in vitro and in vivo revealed that the expression of FoxP2 was significantly higher in the neurons expressing ICS-shFoxP2 compared to those expressing CS-shFoxP2, suggesting that a second AAV expressing Cre-GFP could be used to rescue the knockdown of FoxP2 (Fig. 1c, d and Supplementary Fig. 1d). Upon the introduction of Cre, we observed a switch between mCherry and tagBGP fluorescence both in vivo and in vitro (Fig. 1c and Supplementary Fig. 1a, d, e), providing a reliable way to trace the knockdown and rescue efficiency as well as their coverage areas.

To test the effect of FoxP2 knockdown on vocal behavior, we made bilateral injections of CS-shFoxP2 into Area X of adult birds (159 ± 18 days post hatch (dph), n = 8 birds, Fig. 2a) and monitored their song over two or more months (2.7 ± 0.5 months). Adult zebra finch song is highly stereotyped and characterized by the precise ordering of individual song syllables[19]. We found that knockdown of FoxP2 in adult birds reliably disrupted this stereotypy in as little as three weeks following the viral injections. Changes to song structure included uncharacteristic repetition of individual syllables, replacement and deletion of syllables from the song, and the creation and insertion of entirely new song syllables (Fig. 2d, e, see also Fig. 3e and Supplementary Fig. 2a). Some of these changes gave rise to significant alteration in the syntax of song and large-scale changes in how song bouts were produced (Fig. 2d, e, see also Fig. 3f and Supplementary Fig. 2b).

CS-shFoxP2+ birds appeared to get caught in 'motor loops', repeating a certain song syllable many times before transitioning to the next syllable. Lesions of Area X can cause birds to transiently increase how many times they repeat introductory

notes or other syllables that they already tended to repeat prior to lesions[34,35]. We found significant increases in the repetition of previously repeated syllables at the beginning and/or end of song motifs in 7 out of 8 CS-shFoxP2+ birds (Fig. 2b, d, e). We also found that birds would begin to repeat syllables that were not repeated prior to FoxP2 knockdown (syllable 'd' in Fig. 3e). In addition to increased repetition of introductory notes or pre-existing syllables at the end of the song motif, we found one bird also repeated syllable(s) that were created de novo following knockdown of FoxP2 (syllable 'h' in Supplementary Fig. 2a). In contrast to these changes in song linearity, we did not detect changes in the number of syllable repetitions in any of the CS-shScr+ or ICS-shFoxP2+ birds (Fig. 2a, b). These results suggest that FoxP2 in Area X helps regulate the initiation and termination of vocal sequences and can influence the transition from one syllable to the next within the song motif.

In addition to disrupting song linearity, we found syntax changes in 50% of CS-shFoxP2+ birds (4 of 8 birds, Fig. 2c). These FoxP2 knockdown birds sang songs with disrupted syntax in a substantial proportion of their song bouts (56.2 ± 19.4%), indicating that continued FoxP2 expression in Area X is important for the maintenance of learned syllable sequences and song structure. Most of the syntax changes arose from omitting one or multiple syllables in the middle of the motif following FoxP2 knockdown (Fig. 2d and Supplementary Fig. 2a, b; see also Supplementary Fig. 6a, b). In one bird, we also observed both the dropping of a pre-existing syllable and the addition of a new syllable to the bird's song (Supplementary Fig. 2a, see also Supplementary Fig. 4c). We did not detect these changes in any of the CS-shScr+ or ICS-shFoxP2+ birds which continued to sing their adult song normally after viral injections (0 of 10 birds; Fig. 2c).

Disruptions to song structure observed in our CS-shFoxP2+ birds could be the consequence of a more generalized disruption in song motor control, rather than a selective disruption in control of syllable transitions or syntax. Knockdown of FoxP2 in juvenile birds

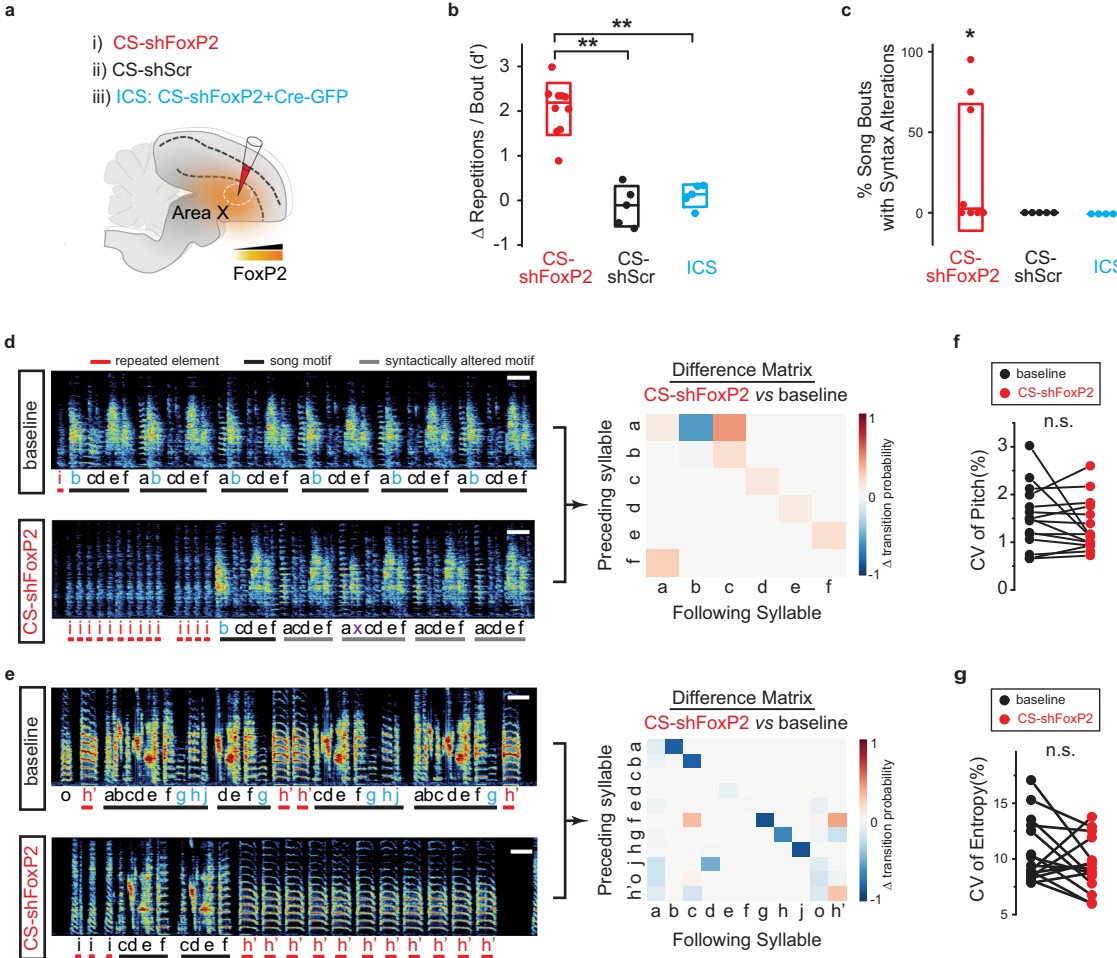

**Fig. 2 FoxP2 Knockdown disrupts syllable sequencing and repetition in adulthood. a** CS constructs were bilaterally injected into Area X of adult birds, alone (i, CS-shFoxP2; ii, CS-shScr) or with Cre-GFP (iii, CS-shFoxP2/Cre-GFP, termed Inverted Cre-Switch (ICS)). **b** Changes in the number of vocal repeats per song bout for CS-shFoxP2+ birds ($d' = 2.05 \pm 0.41$[mean $\pm$ SEM], $n = 8$ birds), CS-shScr+ birds ($d' = -0.13 \pm 0.56$[mean $\pm$ SEM], $n = 5$ birds), and ICS+ birds (blue circles, $d' = 0.11 \pm 0.31$[mean $\pm$ SEM], $n = 5$ birds). Repetition changes in CS-ShFoxP2+ birds were significantly greater than CS-shScr+ and ICS+ birds (2 months post injection vs baseline, CS-shScr, $p = 0.0023$; ICS, $p = 0.015$, Kruskal–Wallis test). Box indicates the median $\pm$ SD. **c** % syntax alterations for CS-shFoxP2+ birds ($n = 8$ birds), CS-shScr+ birds ($n = 5$ birds), and ICS+ birds ($n = 5$ birds) 2 months post injection. Syntax alterations was significantly greater than zero in CS-ShFoxP2+ birds ($p = 0.02$, one tailed one sample $t$ test). Box indicates the median $\pm$ SD. **d** Left panel: song recorded at baseline and 2 months after bilateral injection of CS-shFoxP2 in Area X of an adult bird. The number of repetitions of introductory elements 'i' (red) in each song bout were increased and syllable 'b' (blue) was omitted in a subset of motifs. Each letter indicates an individual syllable. Scale bar, 200 ms. Right panel: difference transition matrices. Subtracting the syllable transition matrix at 2 months following CS-shFoxP2 injection from the matrix at baseline reveals changes in the syllable transitions following FoxP2 knockdown. **e** Left panel: song recorded from a second bird at baseline and 2 months after bilateral injection of CS-shFoxP2 in Area X. The number of repetitions of syllable 'h' (red) increased and other vocal elements (syllables 'g', 'h', and 'j', blue) were omitted. Syllable 'h' is considered a variant of syllable 'h'. Scale bar, 200 ms. Right panel: difference transition matrices. **f** Variability of syllable pitch at baseline (CV = $1.52 \pm 0.17\%$[mean $\pm$ SEM]) and two months post injection of CS-shFoxP2+ birds (CV = $1.34 \pm 0.14\%$[mean $\pm$ SEM]; $p = 0.75$, $n = 15$, Wilcoxon signed-rank test). **g** Variability of syllable entropy at baseline (CV = $10.7 \pm 0.75\%$[mean $\pm$ SEM]) and two months post injection of CS-shFoxP2(CV = $9.46 \pm 0.62\%$[mean $\pm$ SEM]; $p = 0.17$, $n = 15$, Wilcoxon signed-rank test).

is reported to increase the trial-to-trial variability of how individual syllables are produced, and knockdown in adult birds disrupts social modulation of syllable variability[10,11]. To examine if our FoxP2 knockdown approach disrupts trial-to-trial syllable variability, we quantified the coefficient of variation (CV) in syllable pitch and syllable entropy before and 2 months following FoxP2 knockdown. Although we did not test female-directed song, we failed to find disruptions to syllable variability when birds sang alone in their home cage following knockdown (Fig. 2f, g), suggesting that the repetition and syntax changes observed in our experiments did not emerge from a general increase in motor variability. Nonetheless, we observed a decrease in acoustic and sequential similarity between post-knockdown song and pre-knockdown song

(Supplementary Fig. 3a, b). Consistent with our analysis of syllable pitch and entropy, we did not find changes in motif level trial-to-trial acoustic similarity following FoxP2 knockdown, but we did find an increase in the CV of sequential similarity following knockdown, a finding that is in-line with changes in song linearity described above (Supplementary Fig. 3c, d). Together, these results demonstrate an essential and previously unappreciated role for FoxP2 in the selection and sequencing of learned vocal motor actions in adult zebra finches.

**Syllable repetitions are rescued with reversal of FoxP2 knockdown.** Given the large-scale changes to adult song observed

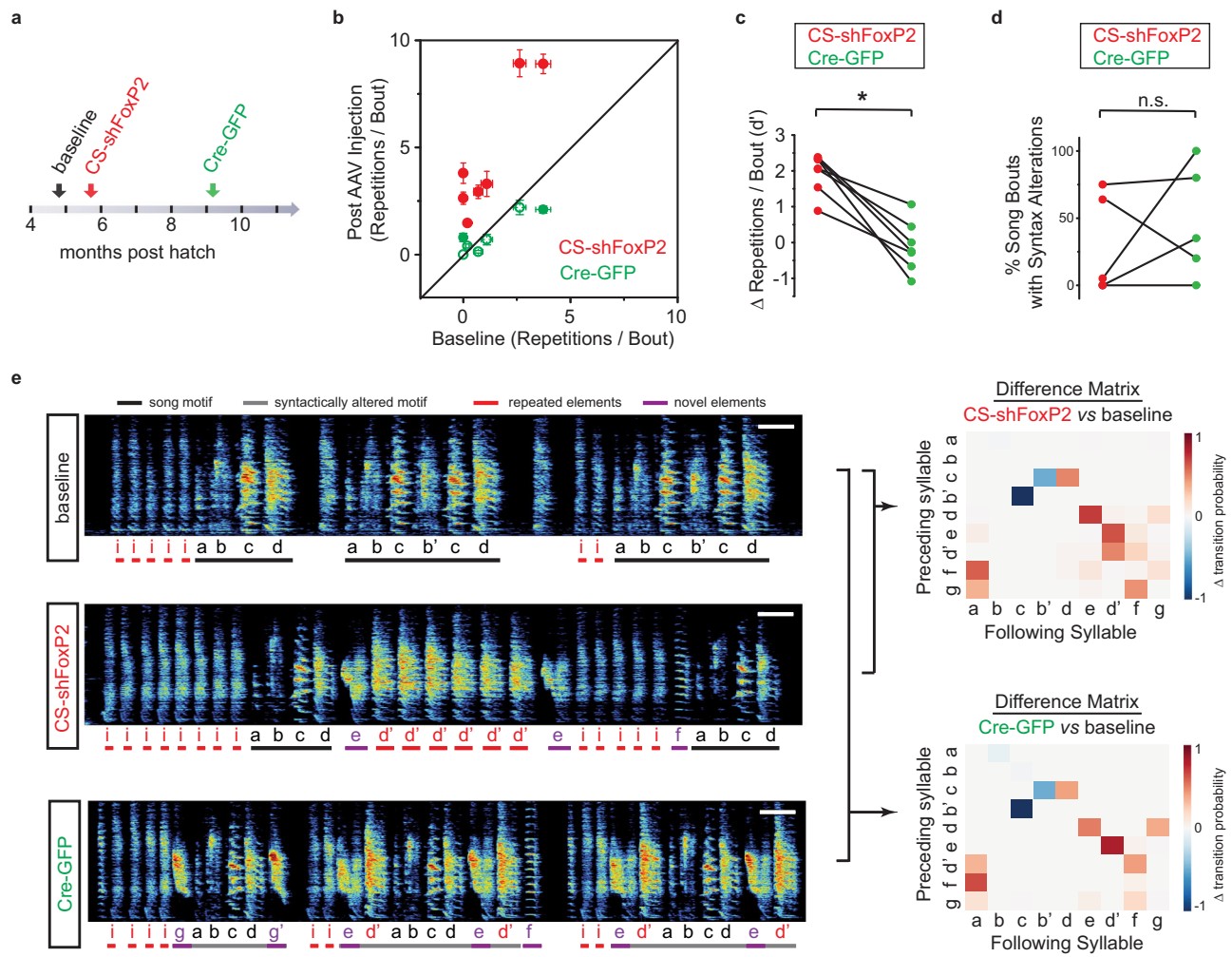

**Fig. 3 Reversal of FoxP2 knockdown in adult zebra finches rescues vocal repetitions but not disruptions in song syntax. a** Strategy for FoxP2 rescue in Area X. **b** Comparison of the number of syllable repetitions per song bout (mean ± SEM) at baseline versus 2 months after injection of CS-shFoxP2 or Cre-GFP ($n = 5$ birds). Red filled circles represent vocal elements with significant differences between baseline and post CS-shFoxP2 injection ($p < 0.0001$ and $p = 0.012$, Kruskal–Wallis test); green open circles, vocal elements with nonsignificant differences ($p > 0.2$, Kruskal–Wallis test); green filled circles, vocal elements with significant differences ($p = 0.014$ and $p = 0.027$, Kruskal–Wallis test) between baseline and post Cre-GFP injection. The number of syllable repetitions after Cre-GFP injection was significantly lower than the number of repetitions of the same syllables after CS-shFoxP2 injection ($p = 0.016$, $n = 7$, Wilcoxon signed-rank test). **c** Changes in the number of repeats per song bout, expressed in units of $d'$, for CS-shFoxP2+ birds ($n = 5$ birds) 2 months following CS-shFxoP2 injection ($d' = 1.94 ± 0.21$[mean ± SEM]) and 3 months following Cre-GFP injection ($d' = -0.011 ± 0.27$[mean ± SEM]). Changes in the number of repetitions relative to baseline were significantly decreased following Cre-GFP injection ($p = 0.016$, Wilcoxon signed-rank test). **d** % syntax alterations for CS-shFoxP2+ birds ($n = 5$ birds) 2 months following CS-shFxoP2 injection and 3 months following Cre-GFP injection. Changes were significantly greater than zero following Cre-GFP injection ($p = 0.033$) but not following CS-shFoxP2 injection ($p = 0.084$, one tailed one sample $t$-test). The frequencies of syntax alterations were not significantly changed following Cre-GFP injection ($p > 0.99$, Wilcoxon signed-rank test). **e** Left panel: spectrograms of song from one bird at baseline, 4 months after CS-shFoxP2 injection, and 3 months after Cre-GFP injection. The number of repetitions of introductory elements 'i' and syllable 'd''(red) gradually increased and novel vocal elements (syllables 'e', 'g', 'g'') emerged. Each letter indicates an individual syllable. Syllables 'b'', 'd' and 'g'' are variants of syllables 'b', 'd', and 'g', respectively. Scale bar, 200 ms. Right panel: difference transition matrices. Subtracting the syllable transition matrix at 4 months following CS-shFoxP2 injection (top) or at 3 months following Cre-GFP injection (bottom) from the matrix at baseline reveals changes in the syllable transitions during reversible knockdown of FoxP2.

following knockdown, we next investigated whether reversal of FoxP2 knockdown would result in recovery of the bird's song. We injected Area X with an AAV expressing Cre-GFP 2–6 months following injection of CS-shFoxP2. AAVs expressing CS-shFoxP2 and Cre-GFP were injected at $177 ± 24$ dph and $277 ± 23$ dph, respectively (Fig. 3a). We found that reversal of FoxP2 knockdown largely eliminated the increased repetition of introductory notes and song syllables within 3 months of Cre-GFP injection (Fig. 3b, c, e and Supplementary Fig. 2a). Unlike FoxP2 knockdown song, the acoustic similarity and sequential similarity of post-rescue song was not significantly different from baseline

(Supplementary Fig. 3e, f). In addition, the trial-to-trial variability of syllable acoustics in post-rescue song was not significantly different from baseline (Supplementary Fig. 3g, h).

In contrast, changes in song syntax accumulated during FoxP2 knockdown were either largely preserved or became more severe (3 of 5 birds) following injection of Cre-GFP (Fig. 3d and Supplementary Figs. 4a, e, 5a, i, 6c, e). While dropping syllables was common following injection of CS-shFoxP2, we found that incorporation of new vocal elements into the song motif was more common during the song recovery period (Fig. 3e and Supplementary Fig. 6a–e). Of the two CS-shFoxP2+ birds who

did not show any syntax changes following injection of CS-shFoxP2, we found that in one, the frequency of syllable insertions per bout increased by 35% following reversal of the knockdown, while the other bird did not show any change in syntax throughout the time course of our experiment (Supplementary Fig. 4b, d). Only one of the five birds undergoing reversal of FoxP2 knockdown showed a decrease in syllable omissions following injection of Cre-GFP (Supplementary Fig. 2d–e, 4d). Together, these results indicate that reversal of FoxP2 knockdown is sufficient to return syllable repetition rates back to those observed prior to FoxP2 knockdown, but also resulted in birds ultimately singing songs that could be different from their pre-knockdown song due to the sustained omission of pre-existing syllables and/or the addition of new syllables into their song.

**Single-nucleus RNA sequencing reveals effects of FoxP2 knockdown**. FoxP2 is broadly expressed in Area X and the surrounding striatum[17] and, as a transcription factor, can have broad effects on gene expression. To identify subpopulation(s) of cells most affected by FoxP2 knockdown and the potential cellular mechanisms for disruptions in song fluency, we performed single-nucleus RNA (snRNA) sequencing on tissue samples of Area X from CS-shScr+ birds and CS-shFoxP2+ birds (for each group, 4 hemispheres total from 2 individuals). We hypothesized that knockdown of FoxP2 would have broad effects on gene expression, based on the extensive knockdown-related disruptions in song. We identified 3388 genes that were significantly differently expressed between the FOXP2+ cells in the scramble and knockdown groups after adjusting for multiple comparisons, for which we then carried out a gene ontology analysis ("Methods"). In line with our hypothesis, many categories of gene function were altered in the knockdown group, such as those involved in cell-cell signaling, neurogenesis, neuron projections, and synapses (Supplementary Fig. 7a). Additionally, we found that 31% of genes denoted as being most strongly linked to autism in the SFARI Gene database were significantly different between FOXP2+ cells in the control and knockdown groups (Supplementary Fig. 7b). Of 366 previously identified downstream targets of FOXP2 in humans, 41 (11.2%) were also differentially expressed between the control and knockdown groups (Supplementary Fig. 7c)[42]. Somewhat surprisingly, these targets were not more likely to be those enriched in the basal ganglia (9/84), and also included targets enriched in the inferior frontal cortex (9/82) and lung (9/82). These results suggest that FoxP2 expression in striatal circuits regulates many signaling pathways that can potentially influence cellular networks involved in the production and control of vocalizations.

To understand the effects of FoxP2 knockdown in MSNs specifically, we first cataloged the cell types present in Area X. A combined clustering analysis indicated that FoxP2 knockdown did not significantly alter the distribution or composition of cell types in Area X (Fig. 4a, Supplementary Fig. 8a), and cells from each group were well-represented within each cluster (Fig. 4b). To define cell types, we focused on an independent clustering analysis of the CS-shScr+ birds (Supplementary Fig. 8b) that identified 23 distinct cell groups (Fig. 4c). Most cells were MSNs (identified by the strong collective expression of *Gad2*, *Ppp1r1b*[43,44], and *FoxP1*[45]), with five MSN clusters comprising 68% of total cells in Area X (Fig. 4d). The pallidal-like cells, identified primarily by the expression of *Penk*[43], existed within one cluster and were far less numerous at only 2.4% of all cells (Fig. 4d). Gene markers for the GPi[44,46] and GPe[47,48] were largely absent from any cluster in the dataset (Supplementary Fig. 9), although the pallidal-like cells here contain some similarities to arkypallidal cells reported in mammals[47–49] (Supplementary Fig. 10). We observed eight distinct clusters of GABAergic neurons, some of which likely correspond to known classes of striatal

interneurons such as *Pvalb*+ interneurons, *Sst*+/*Npy*+/*Nos1*+ interneurons, and *Chat*+ interneurons[43,50–53]. Two clusters expressed the glutamatergic transporter gene *Slc17a6*, which could relate to a glutamatergic cell type that has been reported to exist in Area X[54]. The remaining clusters comprised various glial cell types, such as astrocytes and oligodendrocytes (Fig. 4c).

In the snRNA-seq analysis of the CS-shScr+ birds, we observed that *FoxP2* showed a distinct pattern of expression across the five MSN clusters, with clusters 1, 3, and 5 containing many *FoxP2*+ cells, and clusters 2 and 4 containing mainly *FoxP2*− cells (Fig. 5a). Furthermore, dopamine receptor expression also showed distinct patterns within the MSNs: 36% of the 9672 MSNs were exclusively *Drd1*+ and/or *Drd5*+ (notated here as *Drd1/5*+; *Drd1 is* also known as D1A and *Drd2* is also known as D1B[55,56]), 13% were exclusively *Drd2*+, and an additional 18% expressed varying levels of *Drd1/5* and *Drd2* (Fig. 5b). All five MSN clusters contained cells expressing *Drd1/5*, while only three clusters contained cells expressing *Drd2* (Fig. 5b). Dopamine receptors D1C, D3, and D4 (the latter two belonging to the D2 family) were not expressed at appreciable levels. *FoxP2* frequently co-localized with *Drd1*, but not with *Drd2* (Fig. 5c, d). Of all cells that were *Drd1/5*+ and *Drd2*−, 61% also expressed *FoxP2*, but of cells that were *Drd1/5*- and *Drd2*+, only 21% co-expressed *FoxP2* (Fig. 5d). To interpret this separation of *Drd2*+ cells across clusters, we performed a differential gene expression analysis by grouping the two clusters containing *Drd1*+/*Drd2*− cells into a putative direct-like pathway and the three clusters containing *Drd1*+ cells and *Drd2*+ cells (separate cells, not necessarily co-localizing *Drd1* and *Drd2*) into a putative indirect-like pathway. Canonically, direct pathway MSNs are *Drd1*+ and *FoxP2*+, and project to the GPi while indirect pathway MSNs are *Drd2*+ and *FoxP2*−, and project to the GPe[57–60]. However, MSNs in Area X corresponding to the mammalian direct and indirect pathways have not been previously identified[61]. In our analysis, the direct-like pathway grouping significantly expressed genes that canonically mark direct pathway MSNs, including *FoxP2*, whereas an indirect-like pathway grouping significantly expressed genes that mark indirect pathway MSNs[58,60,62] (Fig. 5e). Although a deeper characterization of the physiological properties and connectivity will be required, analysis of these transcriptomic data suggest that MSNs in Area X may segregate into two broad classes that resemble direct-like (*Drd1*+/*FoxP2*+) and indirect-like pathways (*Drd2*+/*FoxP2*−) (Fig. 5f), as well as a third *Drd1/2*+ class.

Disruptions of FoxP2 expression have previously been shown to result in increased dopamine levels and reduced expression of dopamine receptors in the striatum[11,26]. Using our snRNA sequencing data, we found that knockdown of FoxP2 causes an overall decrease in the *Drd1* to *Drd2* ratio across MSNs in Area X. This change in the *Drd1* to *Drd2* ratio resulted from decreased expression of *Drd1* and *Drd5* in the direct-like pathway MSNs identified above (Fig. 6a, b). Together, these results suggest that knockdown of FoxP2 alters the balance of dopaminergic receptor distributions across the putative direct-like and indirect-like pathways in Area X, which may drive maladaptive repetition of song syllables and/or disrupt song syntax.

**Syllable repetitions are driven by phasic dopamine release in Area X**. Increased dopamine levels and decreased dopamine receptor expression have been associated with impairments in coordinated movements[63,64], including speech. For example, childhood-onset fluency disorder, also referred to as stuttering or stammering, is associated with a hyperdopaminergic tone in the striatum[65–67]. Dopaminergic antagonists have been shown to lessen stuttering and dopaminergic agonists are reported to transiently induce speech disturbances similar to stuttering in

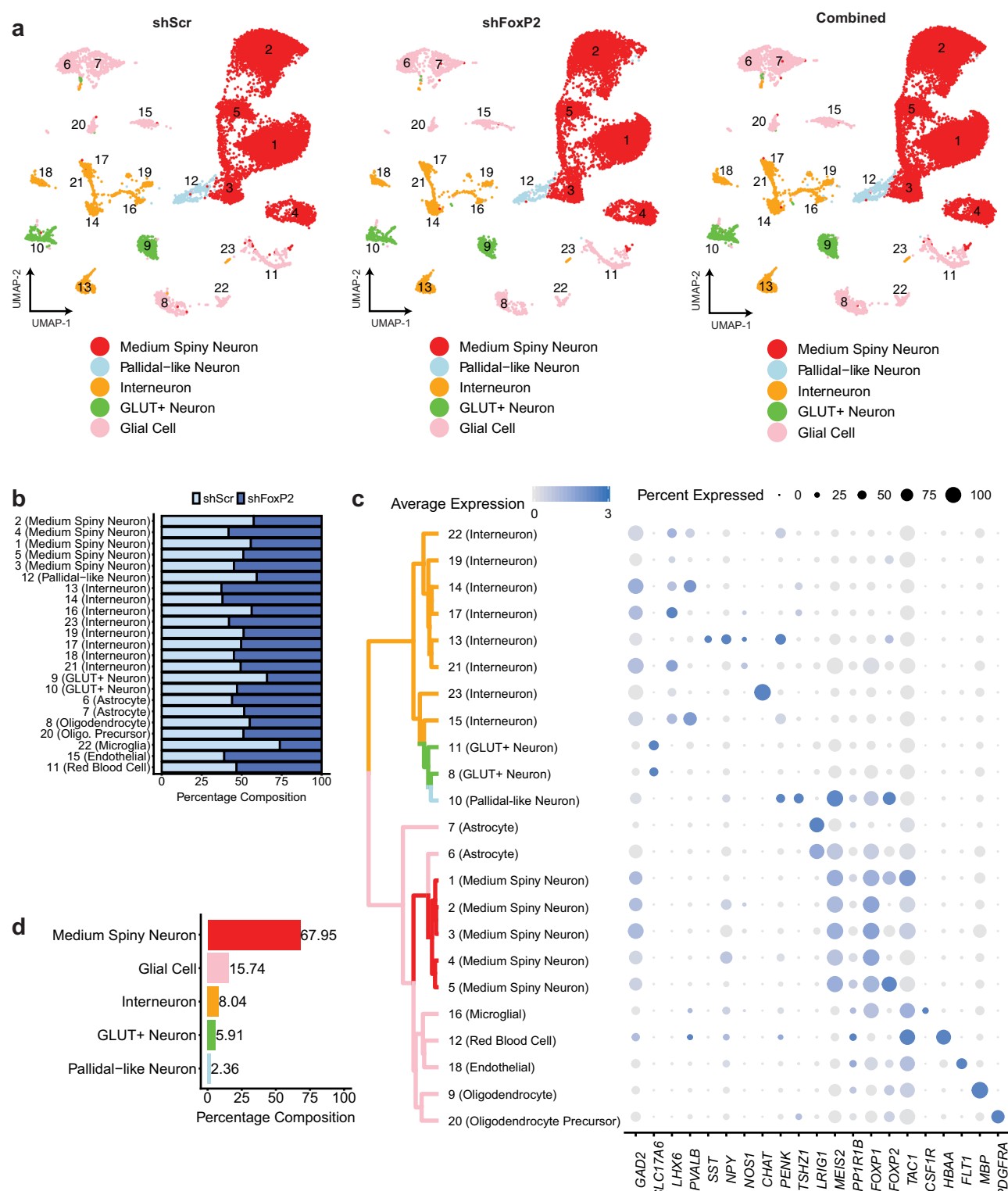

**Fig. 4 Overview of cell types in Area X. a** UMAP projections of nuclei from Area X (left, shScr subset, $n = 14,289$; middle, shFoxP2 subset, $n = 12,956$; right, combined analysis, $n = 27,245$). Clusters are numbered in ascending order by decreasing size (1-largest; 23-smallest). **b** Cluster composition by each dataset (shScr or shFoxP2). **c** Hierarchical clustering and cell type gene marker expression of an independent analysis of the shScr group. For marker gene expression, the size of the dot indicates the percent of nuclei within a cluster expressing a given gene, and the color of the dot indicates the average normalized expression level. **d** Overall cell type composition in Area X of the shScr group.

people with typically fluent speech[68–70]. In addition, mice humanized for Foxp2 have decreased dopamine levels in the striatum, while heterozygous Foxp2 knockout mice have elevated striatal dopamine[26,37]. Given our observed effects following

FoxP2 knockdown in adult birds and the strong ties between FoxP2 expression and dopamine signaling[11,18,26,37], we asked if sustained manipulations of either dopaminergic tone or phasic input would lead to disruptions in syllable repetition or song

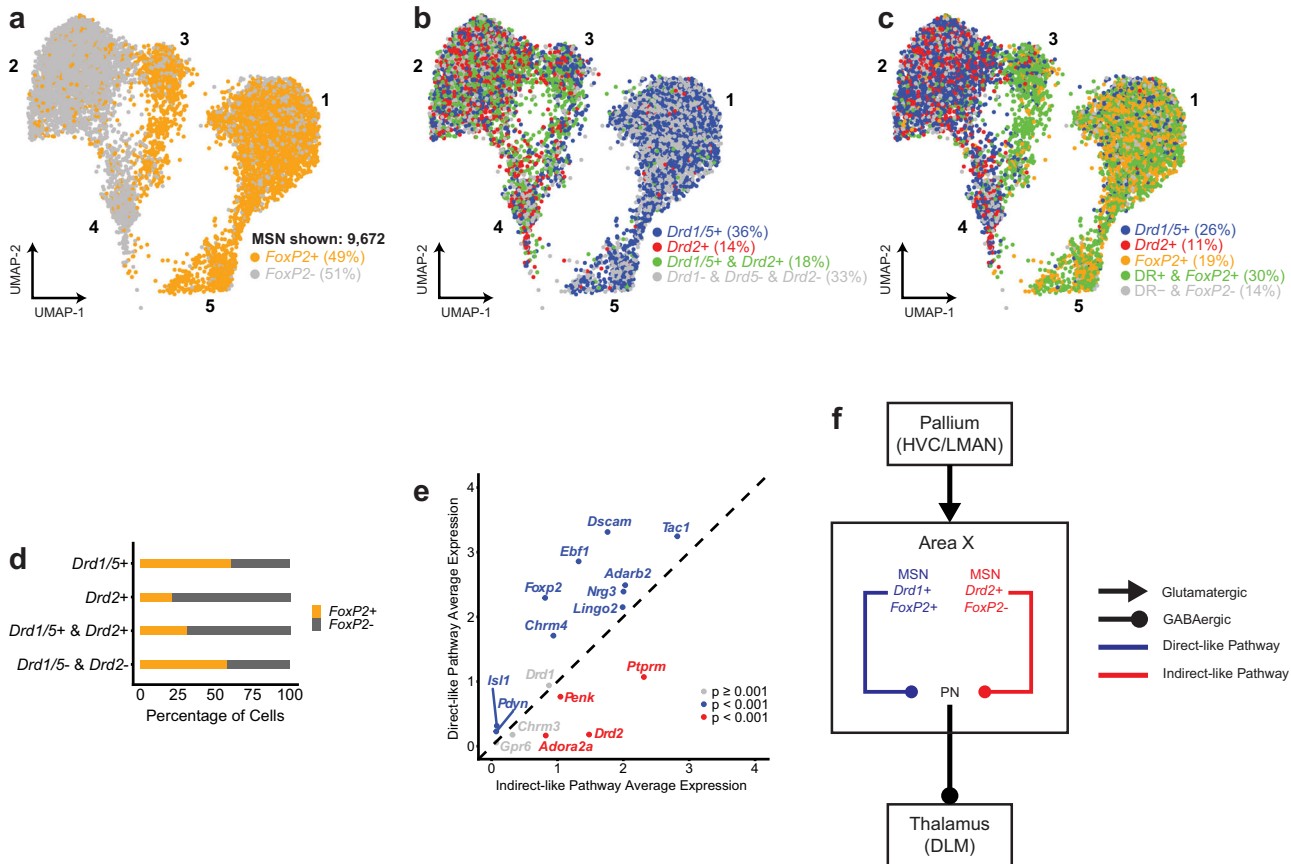

**Fig. 5 Patterns of *FoxP2* and dopamine receptor expression in Area X. a** UMAP projection of MSN clusters. Nuclei are colored based on *Foxp2* expression: *FoxP2+* (orange) or *FoxP2−* (gray). Percentages are rounded and reflect a proportion of the total number of MSNs. **b** UMAP projection of MSN clusters. Nuclei are colored based on the exclusive expression of *Drd1/5* (blue), *Drd2* (red), both (green), or neither (gray). Percentages are rounded and reflect a proportion of the total number of MSNs. **c** UMAP projection of MSN clusters. Nuclei are colored based on the exclusive expression of *Drd1/5* (red), *FoxP2* (orange), any dopamine receptor and *FoxP2* (green), or none (gray). Percentages are rounded and reflect a proportion of the total number of MSNs. **d** A stacked bar plot illustrating the percentage of nuclei expressing *FoxP2*, classified by dopamine receptor expression. **e** A scatter plot showing the differential expression of genes that distinguish nuclei grouped into a putative direct-like pathway (clusters 1 and 5) and indirect-like pathway (clusters 2, 3, and 4). Blue indicates direct-like pathway genes with $p < 0.001$, red indicates indirect-like pathway genes with $p < 0.001$, and gray indicates genes with $p \geq 0.001$. **f** A hypothesized model for the circuitry of MSNs in Area X based on these data.

syntax resembling those observed following FoxP2 knockdown. We directly manipulated dopaminergic inputs to Area X across several days in freely singing birds using pharmacological or optogenetic approaches to test if and how these manipulations resulted in song disruptions, and their similarity to the disruptions observed following FoxP2 knockdown.

First, we tested whether chronic elevation of dopamine could elicit gross changes in song. We implanted reverse microdialysis probes bilaterally in Area X and individually infused dopamine, D1-like agonists, and D2-like agonists, each for several days when birds were singing alone (Supplementary Fig. 11a–c). Remarkably, and in contrast to the vocal deficits observed following FoxP2 knockdown, birds continued to sing at normal rates and without disruptions in the song's spectral structure, syntax, or syllable repetition during direct infusion of dopamine or dopamine receptor agonists (Supplementary Fig. 11d).

Next, we tested if sustained manipulations to phasic dopamine activity cause disruptions to song sequences. We systematically tested the effect of both phasic increases and phasic decreases in dopamine release over several days. AAVs expressing an axon-targeted channelrhodopsin (ChR2), archaerhodopsin (ArchT), or green fluorescent protein (GFP) were injected bilaterally into VTA, and birds were implanted with optical fibers over Area X (Fig. 7a, b). Bilateral optical illumination of VTA axon terminals

in Area X was targeted to an individual syllable in the song motif for 3–12 consecutive days (Fig. 7c and Supplementary Fig. 12a). Light pulse delivery was contingent on natural trial-to-trial variation in the pitch of the targeted syllable. In agreement with our previous results using this method[38], we found that optogenetic excitation and inhibition elicited learned changes in the pitch of the targeted syllable on future performances (Fig. 7d). In addition to changes in the pitch of the targeted syllable, we found that phasic increases in dopamine also resulted in a significant increase in syllable repetitions at the beginning and/or end of song motifs, similar to those changes we observed in birds in which FoxP2 expression was knocked down in Area X (Figs. 7e–g, 8a, and Supplementary Fig. 12b). This increase in syllable repetitions was observed in all ChR2 expressing birds by the third day of phasic stimulation. We did not observe disruptions in the selection and sequencing of syllables in birds that received phasic inhibition of dopamine release during singing, or in birds expressing GFP (Figs. 7d, 8b, c, and Supplementary Fig. 12c, d).

The increased repetition of song syllables generalized to all song performances, not just optically stimulated trials, and persisted for two or more days after optical stimulations were discontinued (Fig. 8a and d, e). This suggests that phasic increases in dopamine do not have a direct influence on ongoing vocal motor actions but

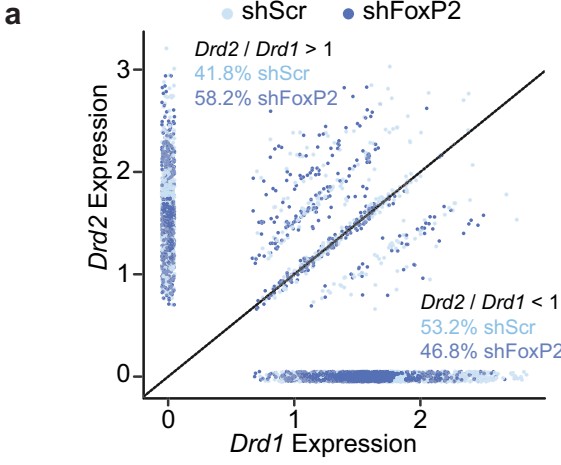

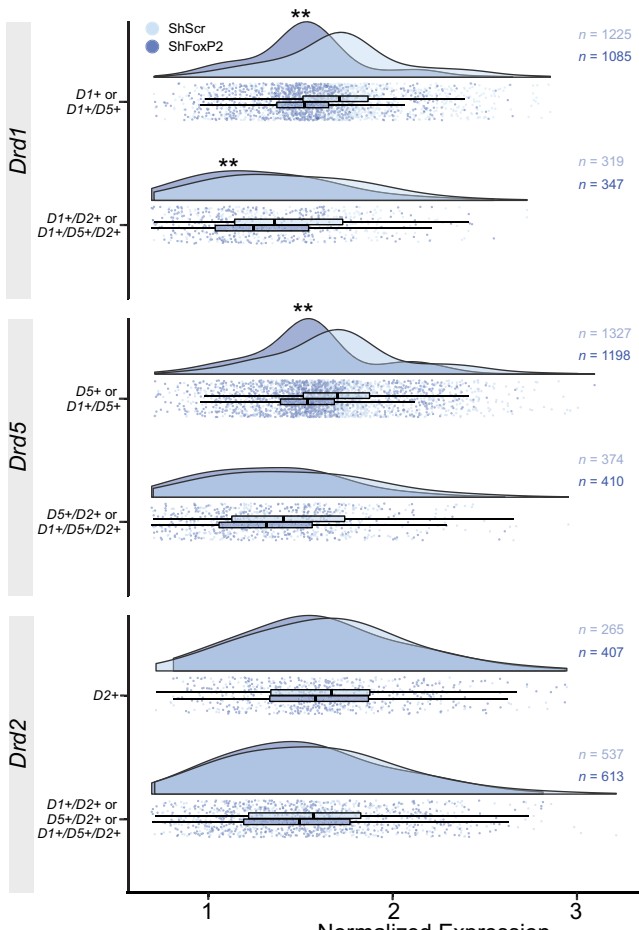

**Fig. 6 Altered expression of dopamine receptors resulting from FoxP2 knockdown. a** A scatterplot illustrating the correlation between normalized expression of *Drd1* and *Drd2* in *FoxP2+* striatal cells that are also *Drd1+* or *Drd2+*. Each point represents a nucleus. Diagonal line indicates the line of equality. Cells above the line have a ratio of *Drd2/Drd1* greater than 1. Cells below the line have a ratio of *Drd2/Drd1* below 1. **b** Distribution and boxplots of normalized expression for *Drd1*, *Drd5*, and *Drd2* in *FoxP2+* MSNs in CS-shScr+ and CS-shFoxP2+ birds. Cells are grouped by the expression of dopamine receptors, e.g., *D1+* indicates the cells express *Drd1* only and no other dopamine receptor, whereas *D1+/D5+* indicates the cells express both *Drd1* and *Drd5*. The expression level of *Drd1* in cells from CS-shFoxP2+ birds was significantly lower than in CS-shScr+ birds ($p < 0.001$, Welch's *t*-test) in all *Drd1+* cells, as indicated by the leftward shift of the distribution of normalized expression in the CS-shScr+ population. The expression of *Drd2* in *FoxP2+* MSNs did not differ between the two groups, as no shift in the distribution of normalized expression is seen between the populations. The lower and upper bounds of the boxes indicate the 25th and 75th percentiles; the whiskers extend in either direction from the bound to the furthest value within 1.5 times the interquartile range.

of the sixth syllable (Fig. 7g1) resulted in vocal repetitions at the end of the song motif, while optogenetic excitation of the third syllable (Fig. 7g2) resulted in vocal repetitions at the beginning of the song motif.

Notably, optogenetic stimulation of VTA terminals did not recapitulate the full range of vocal disruptions observed following FoxP2 knockdown. We did not observe dropping of syllables or creation of new syllables following optogenetic stimulation. Together, these results suggest a common role for phasic dopamine signaling and FoxP2 expression in regulating precise sequencing of vocalizations, as disruptions in these physiological circuits lead to maladaptive repetition of song syllables.

Since disruptions in vocal sequencing emerge while birds are also learning to change the pitch of the optically targeted song syllable, we examined whether there was a relationship between adaptive (pitch learning) and maladaptive (syllable repetitions) forms of vocal plasticity. We found that optical inhibition experiments drove changes in the pitch of song syllables comparable to those seen in optical excitation experiments, yet these manipulations did not result in maladaptive vocal repetitions (Fig. 7d, e). This suggests that reinforcement-based learning of changes in pitch can occur independent of maladaptive changes in syllable sequencing. In addition, we found that the recovery from optogenetic-induced changes in vocal repetitions occurred on timescales that differed from recovery in pitch learning (Fig. 8d, e). Recovery trajectories for these two types of learning varied from bird to bird, but their decoupling further suggests that reinforcement-based changes in how syllables are sung (pitch learning) occurs independent of maladaptive changes to circuits that may help regulate vocal sequences.

Together, these findings indicate that increased phasic excitation of dopaminergic VTA terminals in Area X causes problems with initiating and terminating song, marked by birds repeating syllables at the beginning and ending of song. That only phasic excitation is sufficient to drive these disruptions raises the possibility that the putative direct-like pathway through the striatum, and FoxP2 expressing neurons in this circuit, functions in fluently starting and stopping zebra finch song.

## Discussion
Precise sequencing of vocal motor actions is necessary for vocal communication. While recent studies have begun to clarify the role of FoxP2 and reinforcement signals in learning how to properly produce individual syllables[11,18,27,28,38,71–73], the identities of cells

may maladaptively influence the selection and sequencing of vocal motor actions. Consistent with this interpretation the bird only started to repeat the optically targeted syllable in one case (1 of 8 birds) (Supplementary Fig. 12b). In this case, the bird began to repeat a pair of syllables in the middle of its song motif, suggesting that vocal repetitions could emerge at any point in song and are not necessarily confined to the beginning or ending of the song motif. We also found that which syllable was optically stimulated appears to influence which other syllable(s) birds began to repeat (Fig. 7g). In a bird with a six-syllable motif, optogenetic excitation

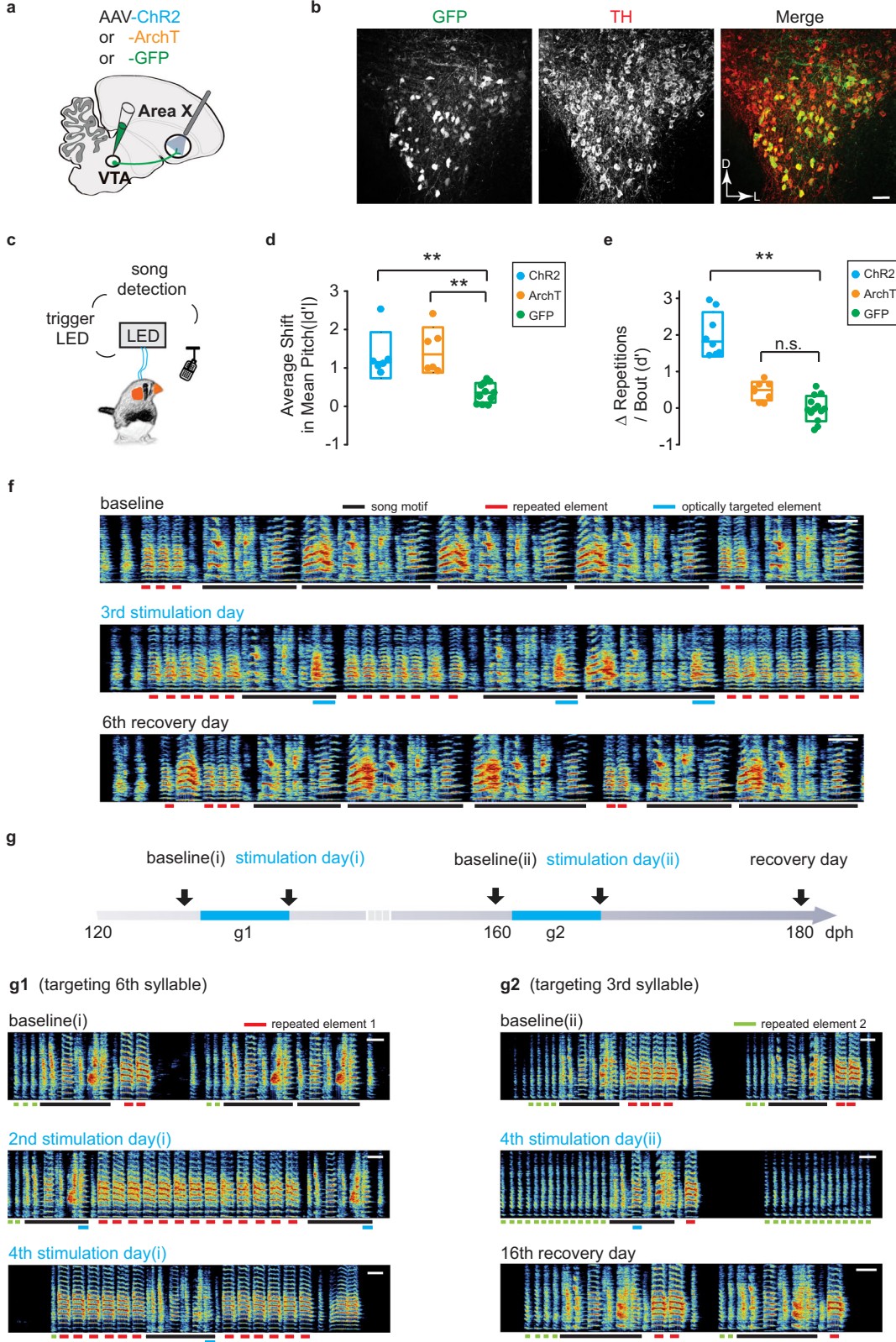

and circuits that control the selection and sequencing of syllables have remained unclear. We show that the expression of FoxP2 is critical for the maintenance of adult vocalizations and that knockdown of its expression causes syllable repetitions and disruptions to song syntax. Restoring FoxP2 expression later in adulthood resulted in recovery from aberrant vocal repetitions in all cases while alterations in syntax were largely maintained. This adult plasticity is particularly striking given that birds of this age are thought to have limited song plasticity and to be less reliant on auditory feedback to maintain their songs[74]. We identify phasic dopamine as a selective mediator of maladaptive changes in song sequences and, therefore, show that dopamine not only functions in reinforcement-based learning of song syllables[38] but also in the accurate sequencing of syllables in adulthood (Fig. 9).

**Fig. 7 Song contingent excitation of dopamine terminals in Area X induces dysfluent repetition of song syllables. a** Schematic of experimental design for optogenetic manipulation of dopamine release from VTA terminals in Area X. **b** Representative coronal section through VTA showing that most neurons infected with optogenetic constructs are TH-positive and located in the ventral and ventrolateral portions of VTA. Scale bar, 50 μm. **c** Schematic of closed-loop optogenetic experimental paradigm. **d** Average shift in mean pitch, expressed in units of $|d'|$, for ChR2+ birds ($|d'| = 1.33 \pm 0.6$[mean ± SEM], $n = 6$ birds), ArchT+ birds ($|d'| = 1.47 \pm 0.59$[mean ± SEM], $n = 6$ birds), and GFP+ birds ($|d'| = 0.35 \pm 0.077$[mean ± SEM], $n = 6$ birds). Average shift in mean pitch for both ChR2+ and ArchT+ birds were higher than 0.75, and also significantly higher than control GFP+ birds (ChR2+, $p = 0.0025$; ArchT+, $p = 0.0025$; Kruskal–Wallis test). Box indicates the median ± SD. **e** Changes in the number of repetitions of vocal element per song bout between the baseline day and last illumination day, expressed in units of $d'$, for ChR2+ birds ($d' = 2.02 \pm 0.5$[mean ± SEM], $n = 6$ birds), ArchT+ birds ($d' = 0.46 \pm 0.21$[mean ± SEM], $n = 6$ birds), and GFP+ birds ($d' = -0.015 \pm 0.2$[mean ± SEM], $n = 4$ birds). Change in the number of repetitions of vocal elements in ChR2+ birds were significantly greater than change in GFP+ birds ($p < 0.0001$, Kruskal–Wallis test), but there was no significant difference between ArchT+ and GFP+ birds ($p = 0.12$, Kruskal–Wallis test). Box indicates the median ± SD. **f** Spectrograms of song recorded from a ChR2+ bird at baseline, 3rd stimulation day and 6th recovery day. Light pulses (~455 nm, 100 ms) were delivered over the target syllable during lower pitch variants but not during higher pitch variants. **g** Schematic of the experiment in a ChR2+ bird in which light pulses (~455 nm, 100 ms) were delivered over two different target syllables at different times over the course of 2 months. The bird starts repeating a song element either at the end of its motif (g1) or the beginning of its motif (g2) or both (g1 bottom), depending on which syllable in the song was optogenetically targeted. Scale bar, 200 ms.

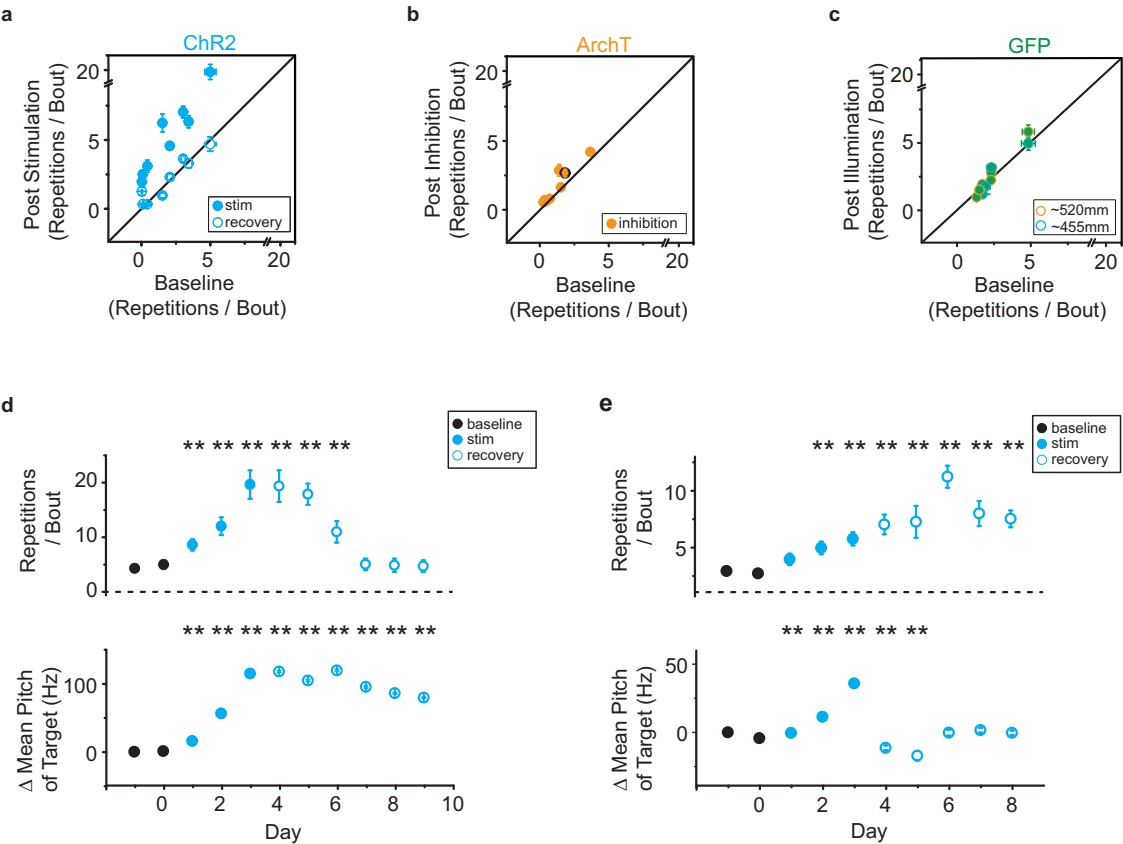

**Fig. 8 Optical stimulation causes vocal repetitions independent of changes in pitch. a** Comparison of the number of repetitions of vocal element per song bout (mean ± SEM) for the baseline day versus last stimulation day in 6 ChR2+ birds (filled, $p < 0.0001$, Kruskal–Wallis test) or last recovery day (open, $p > 0.35$, Kruskal–Wallis test, $n = 8$ vocal elements). The number of repetitions of vocal elements on the last stimulation day was significantly higher than the number of repetitions on either the baseline day or the last recovery day ($p = 0.0081$, Friedman test), with no significant difference between the baseline day and the last recovery day ($p > 0.99$, Friedman test). **b** Comparison of the number of repetitions of vocal element per song bout (mean ± SEM) on the baseline day versus last inhibition day in 6 ArchT+ birds (5 filled circles, $p > 0.08$; 1 filled with black outline, $p = 0.028$, Kruskal–Wallis test, $n = 8$ syllables). **c** Same as **b**, but for GFP+ birds (blue outline, $p > 0.8$, Kruskal–Wallis test, $n = 6$ syllables from 3 birds; orange outline, $p > 0.15$, Kruskal–Wallis test, $n = 7$ syllables from 3 birds). Blue and orange outlines indicate birds illuminated by LED with wavelength of ~455 nm or 520 nm, respectively. **d** The number of syllable repetitions per song bout (top, mean ± 95% confidence interval[CI]) for the bird shown in Fig. 7f and changes of the mean pitch of the optically targeted syllable(bottom, mean ± 95% CI)(blue line in Fig. 7f) during baseline (black circles, days −1 and 0), stimulation (filled blue circles, days 1–3), and recovery (open blue circles, days 4-9). The number of repetitions of the affected syllable (red line in Fig. 7f) per song bout was significantly increased during days 1–6 relative to baseline ($p < 0.01$, Kruskal–Wallis test), whereas changes in the mean pitch of the target syllable during days 1–9 were significantly higher than changes at baseline ($p < 0.01$, Kruskal–Wallis test). **e** same as **d**, but for another bird. The number of repetitions of the affected syllable per song bout was significantly increased during days 2–8 relative to baseline ($p < 0.01$, Kruskal–Wallis test), whereas changes in the mean pitch of the target syllable during days 1–5 were significantly higher than changes at baseline ($p < 0.01$, Kruskal–Wallis test).

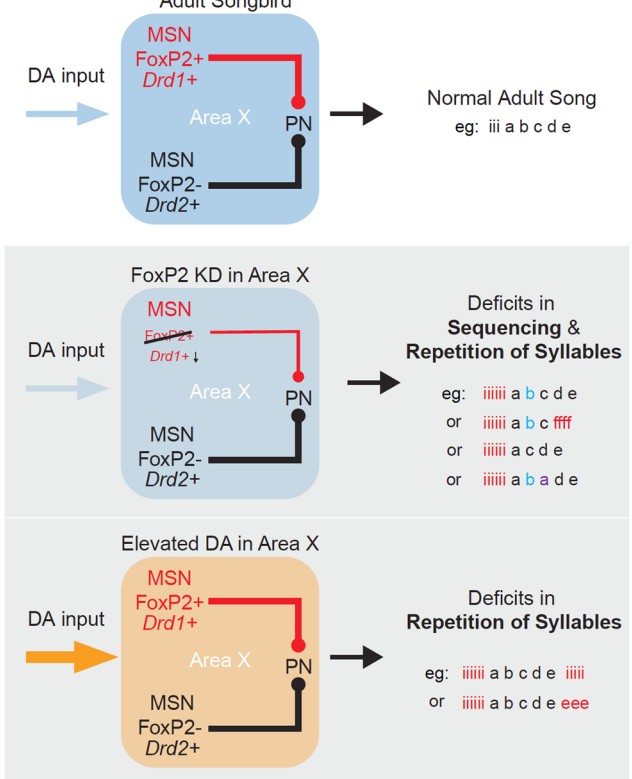

**Fig. 9 Hypothetical role of Area X in adult vocalization.** Coordinated activity between direct- and indirect-like pathways, as well as appropriate dopaminergic (DA) input into Area X, is needed for the production of normal adult song. Reduced FoxP2 expression in Area X may cause an imbalance in activity between the direct- and indirect-like pathways and result in deficits in song sequencing and the repetition of syllables. Similarly, elevated DA input into Area X may introduce the repetition of syllables at the onset or offset of song.

Heterozygous mutations of *FOXP2* cause Childhood Apraxia of Speech, also referred to as Developmental Verbal Dyspraxia[31]. This speech impairment is thought to result in part from disruptions in developmental plasticity of basal ganglia circuits[75]. The role of FOXP2 in the maintenance of adult speech is not known. Altering the expression of FoxP2 in Area X impairs song imitation in juvenile birds[10], but it had not been shown to be necessary for maintenance of the adult song. A previous study of FoxP2 knockdown in adult Area X focused on changes in context-dependent syllable variability[11]. Using the same hairpin sequence used in previous studies[10,11] but expressing it in Area X using a novel AAV CS-shFoxP2 construct, rather than lentiviral based methods, we show that FoxP2 expression is necessary for maintenance of adult song sequences and syntax. Although we did not observe changes in syllable variability following FoxP2 knockdown, we did not measure social context-dependent singing in this study. We made this choice because female-directed singing has been shown to elevate dopamine in Area X as well as increase the number of introductory notes that birds sing[19,76–78], variables that we wished to manipulate and quantify, respectively, in the undirected singing condition. We suspect that the broader neural infection in Area X afforded by AAV over lentivirus could account for the stronger effects on song performance and maintenance reported here[79]. Knocking down FoxP2 expression in Area X of adult zebra finches drove a significant increase in the repetition of song syllables, the elimination of certain syllables from the song, and the improvisation of new syllables. However,

we did not observe spectral degradation of individual song syllables, as seen following deafening[74,80,81]. This indicates that the ability to control fine-scale features of song was largely undisturbed, while the selection and sequencing of syllables was impaired.

When we rescued FoxP2 knockdown in adult birds, we found that the aberrant repetition of song syllables was eliminated in all birds within ~3 months. This suggests that the progressive and maladaptive changes in behavior driven by disruptions of FoxP2 can in part be overcome with restoration of gene expression. Despite its name, Childhood Apraxia of Speech is a lifelong condition. Finding that birds can recover key aspects of song behavior well into adulthood, a timepoint when song is thought to be mostly rote[74], suggests the possibility that genetic therapies for speech disorders may potentially have relevance even beyond early developmental windows when speech is first learned.

We demonstrate that FoxP2 knockdown and phasic activation of VTA terminals in Area X similarly disrupt the fluent initiation and termination of adult song by increasing the repetition of song syllables. To better understand the relationship between FoxP2 knockdown and dopamine signaling, we used snRNA sequencing to map the cell types in Area X and the gene-expression changes induced by FoxP2 knockdown. We identified populations of *Drd1*+/*FoxP2*+ MSNs and *Drd2*+/*FoxP2*− MSNs in Area X, consistent with the molecular profiles of striatal direct and indirect MSN populations seen across other vertebrates[59]. Nearly 20% of the MSNs in Area X co-express *Drd1* and *Drd2*. While this proportion is less than what has been previously reported[55,82], it is much higher than what has been reported in the mammalian striatum, where MSNs co-expressing *Drd1* and *Drd2* only represent about 1–4% of all MSNs[44,58,60], although these previous studies did not examine *Drd5*. The broad groupings of MSNs identified here highlight commonalities in the gene markers between Area X and mammalian MSN populations[58,60,62]; however, this is not the case for all genes. While *Drd2* and *FoxP2* are differentially expressed across MSN neuronal clusters, *Drd1* is not. Furthermore, a substantial portion (33%) of the MSNs in our data set do not express any dopamine receptor, a much higher percentage than what has been reported in mammalian striatum[44,58,60]. This may reflect the inability to detect transcripts expressed at low levels in our data, or evolutionary differences in the composition of Area X and portions of the mammalian striatum that have been studied to date. Consistent with a previous study[11], we also found diminished expression of *Drd1*+ and *Drd5*+ in Area X following FoxP2 knockdown (Fig. 6).

The excess dopamine hypothesis for stuttering proposes that hyperdopaminergic tone in the basal ganglia causes speech dysfluencies, including maladaptive vocal repetitions and difficulty initiating speech[65–67]. Interestingly, heterozygous *Foxp2* mutant mice have elevated striatal dopamine[37] and deficits in syllable sequencing, which are suggested to be similar to sequencing deficits in humans with *FOXP2* mutations/Childhood Apraxia of Speech[32,33]. Our finding that phasic activation of dopamine release in Area X and knockdown of FoxP2 cause similar maladaptive repetition of song syllables helps link these findings and indicates that common disruptions in circuit function may underlie seemingly disparate disorders, like stuttering and Childhood Apraxia of Speech. That we only observed disruptions in singing following song-contingent phasic activation of VTA terminals, and not during chronic infusion of dopamine or dopamine receptor agonists, further suggests a strong behavior or vocal sequence specific contingency in how elevated dopamine influences vocal fluency. It is common in stuttering to have particular words or sounds that different individuals find especially difficult to produce. Understanding if or how dopamine is dynamically regulated during production of these and other less

problematic words or sounds may be particularly revealing to the underlying circuit disruptions associated the stuttering.

When considered within the context of the broader literature, it appears that Area X plays a dual role in song. It is involved in learning how individual song syllables should be sung and in controlling larger scale selection and sequencing of these syllables[13–15,34–36]. Positive and negative dopaminergic reinforcement signals guide how individual song syllables are sung on future performances[28,38], while disruptions to the contributions of the putative direct-like and indirect-like pathways may regulate syllable selection and sequencing (current results). Knockdown of FoxP2 leads to a decrease in $Drd1/Drd2$ ratios, which may drive an imbalance in direct-like and indirect-like pathways. However, it remains to be determined whether identified behavioral abnormalities are direct consequences of imbalances in inhibitory/excitatory circuits or direct/indirect-like pathways[83]. Continued phasic excitation of dopaminergic inputs, which may preferentially influence activity in the direct pathway[84], consistently resulted in birds having prominent sequence disruptions/repetitions at the beginning and end of their songs. Similarly, expression of the mutant gene fragment that causes Huntington's disease in Area X also causes disruptions in song syllable selection and repetition[36]. These disruptions, however, tend to be more restricted to changes in the middle of the motif and do not accumulate at the initiation and termination of song. Indirect pathway neurons are particularly vulnerable at early stages of Huntington's disease[64,85–87], which together with our findings, raises speculation that disruptions in the putative direct-like pathway could more readily cause vocal repetitions at initiation and termination of vocal motor sequences, whereas disruptions in the putative indirect-like pathway could tend to disrupt sequences in the middle of song. Hierarchical representations of song sequences may therefore rely critically on coordinated activity between different pathways traversing Area X and the precise timing signals that facilitate transitions between individual syllables. The molecular cataloging of Area X cell types, and the tools for reversible genetic manipulations described here, provide the means to start testing these and related hypotheses about the selection and sequencing of vocal motor actions.

Many speech disorders arise from problems in translating speech plans into accurate motor actions[2,88–90] and have been linked to hyperdopaminergic signaling in the striatum[65–67,91]. Together, our findings in Area X indicate that vocal dysfluencies in songbirds may be associated with hyperdopaminergic signaling in the striatum, as seen in human vocal dysfluencies.

## Methods

**Animals**. All experiments were performed on adult male zebra finches (*Taeniopygia guttata*) raised in a breeding facility at UT Southwestern and housed with their parents until at least 50 days of age. During experiments, birds were housed individually in sound-attenuating recording chambers (Med associates) on a 12/12 h day/night cycle and were given ad libitum access to food and water. All procedures were performed in accordance with established protocols approved by the UT Southwestern Medical Center Animal Care and Use Committee.

**Plasmid construction and viral vectors**. The backbone of CS constructs was based on pAAV-EF1α-DO-mCherry (Addgene, #37119)[39], and the fluorescent protein cDNA for tagBFP was cloned from pdCas9::BFP-humanized (Addgene, #44247). Two hairpins (shFoxP2a, target sequence AACAGGAAGCCCAACGTTAG T[10], and shFoxP2i, target sequence ACTCATCATTCCATAGTGAAT) were inserted downstream of the U6 promoter at the base of the Mir-30 stem-loop. The scrambled hairpin (shScr, sequence CCACTGTACTATCTATAACAT) was designed as a control. Hairpins were then assembled into the pTripZ vector (Thermo Scientific, MA, USA) by directional ligation into the XhoI–EcoRI cloning sites. We then replaced the EF1α promoter with a CAG promoter and assembled hairpins together with pTripZ context sequence (between the BspD1 and MluI cloning sites) in the forward orientation, with the tagBFP transgene in the reversed orientation downstream of the mCherry transgene. To validate knockdown efficiency, a zebra finch FoxP2 cDNA clone provided by Erich Jarvis (Rockefeller University) was subcloned into pLenti6.4 using a gateway reaction, adding a V5 tag, to achieve overexpression of FoxP2 in vitro. HEK293T stable cell lines expressing vehicle or Cre-GFP were generated to validate CS constructs in vitro. All N Terminal sites included a Kozak sequence (GCCACC) directly preceding the start codon. Sequence confirmation was done by the McDermott Center Sequencing Core at UT Southwestern Medical Center. The recombinant AAV vectors were amplified by recombination deficient bacteria, One Shot Stbl3 (C737303, Invitrogen, CA, USA), serotyped with AAV1 coat proteins and produced by the University of North Carolina vector core facility (Chapel Hill, NC, USA) with titer exceeding $10^{12}$ vg/ml, the Duke viral vector facility (Durham, NC, USA), IDDRC Neuroconnectivity Core in Baylor College of Medicine (Huston, TX, USA), or in the Roberts lab with titer exceeding $10^{11}$ vg/ml. All viral vectors were aliquoted and stored at –80 °C until use.

**Stereotaxic surgery**. All surgical procedures were performed under aseptic conditions. Birds were anesthetized using isoflurane inhalation (1.5–2%) and placed in a stereotaxic apparatus. Viral injections and cannula or microdialysis probe implantation were performed according to previously described procedures[38]. The approximate stereotaxic coordinates relative to interaural zero and the brain surface were (rostral, lateral, depth, in mm): Ov (2.8, 1.0, 4.75), the center of Ov was located and mapped based on its robust white noise responses; VTA relative to the center of Ov (+0.3, −0.2, +1.8); Area X (5.1, 1.6, 3.3) with 43-degree head angle or (5.7, 1.6, 3) with 20-degree head angle, the boundary of Area X was verified using extracellular electrophysiological recordings. 0.7–2 μl AAVs were injected according to the titer of constructs and allowed 3–8 weeks for expression before birds were involved in behavioral tests, immunohistochemistry, and/or sequencing experiments.

### Behavioral assays

*Song recording*. Acoustic signals were recorded continuously by a microphone (Shure BETA 98A/C) immediately adjacent to the bird's cage using Sound Analysis Pro2011[92] and bandpass filtered between 0.3 and 10 kHz. All songs presented in this paper and used for analysis were recorded when the male was alone in a sound-attenuating chamber.

*Optogenetic manipulation of VTA axon terminals in behaving birds*. All procedures were performed as reported previously[38]. Briefly, male birds were randomly assigned bilateral injection of either AAV-ChR2, -ArchT or GFP constructs in VTA at ~70 dph. Birds were implanted with fiber optics when they were >100 dph. Birds were given at least 1 week to recover from cannula implantation and to habituate to singing with attached optical fibers. Custom LabView software (National Instruments) was used for online detection of preselected target syllables and implementation of closed-loop optogenetic manipulation[93]. 100-ms light pulses were delivered over a subset of variants of the target syllables in real time (system delay less than 25 ms) for 3–12 consecutive days, as described previously[38]. Investigators were not blinded to allocation of optogenetic experiments. 3–5 mW of ~455 nm and 1.5–4 mW of ~520 nm LED output was delivered from the tip of the probe (200 or 250 μm, NA = 0.66, Prizmatix, Israel) to ChR2+ and ArchT+ birds, respectively, while either ~455 nm or ~520 nm light pulses were delivered to GFP+ birds (n = 4). All ChR2+ (n = 6) and ArchT+ (n = 6) birds exhibited significant changes in the pitch of target syllables ($|d'| > 0.75$) and were included in subsequent behavioral analyses.

*Pharmacological Manipulation of DA Circuit in Behaving Birds*. We used two microdialysis systems to chronically infuse dopamine hydrochloride (DA), SKF 38393 hydrobromide (SKF), or (−)-Quinpirole hydrochloride (Qui) (#3548, #0922, and #106, Tocris, MN, USA) into Area X to simulate tonically elevated DA levels. Two male adult birds were implanted bilaterally with probes constructed in house from plastic tubing (427405, BD Intramedic, PA, USA; 27223 and 30006, Micro-Lumen, FL, USA) which served as a drug reservoir, fitted at the end with a 0.7 mm-long semipermeable membrane (132294, Spectra/Por, MA, USA) allowing drug to slowly diffuse into the brain throughout the day[94,95]. Freshly prepared DA, SKF or Qui (400 mM) were used to fill and refilled microdialysis probes every morning for 4 consecutive days following 3 days of dialysis with PBS. Two other birds were implanted with guide cannulas (8010684, CMA, MA, USA) bilaterally over Area X, and microdialysis probes (1 mm membrane length, 6 kDa cutoff, P000082, CMA, MA, USA) were not inserted until birds recovered from surgery and were singing for 2–3 days as described previously[96,97]. Fresh prepared DA, SKF or Qui (100–400 mM), or PBS (for baseline) were continuously delivered to Area X for 3–8 consecutive days at a rate of 0.2 μl/min via a fluid commutator connected to a syringe pump outside the bird's isolation chamber. In all cases birds could freely move and sing during infusion.

### Behavioral analysis

*Song structure*. Zebra finch song can be classified into three levels of organization: syllables, which are individual song elements separated by short silent gaps >5 ms in duration; motifs, which are stereotyped sequences of syllables (demarcated by black lines under spectrograms); and song bouts, which are defined as periods of singing comprised of introductory elements followed by one or more repeats of the song motif with inter-motif intervals <500 ms[77,98].

*Quantification of vocal repetition*. The vocal element being repeated consisted of either a single syllable (e.g., introductory element repeated in Fig. 2d or ending song syllable repeated in Fig. 2e) or multiple syllables (e.g., song syllables repeated in Supplementary Fig. 12b). The number of repetitions of individual vocal element per song bout (n) is defined as the total number of consecutively repeated vocal elements, not including the first rendition each time the element is sequentially produced within the song bout. $d'$ scores were computed to express the changes in the mean number of repetitions of individual vocal element per song bout (n) relative to the last baseline day[38]:

$$d'_i = \frac{\sqrt{2}(n_i - n_b)}{\sqrt{\sigma_i^2 + \sigma_b^2}}$$

$n_i$ is the mean number of repetitions of individual vocal element per song bout on day i and $\sigma_i^2$ is the variance on day i. Subscript b refers to last baseline day. In the case of equal variances ($\sigma_i^2 = \sigma_b^2$), $d'_i$ reports the changes in average repeats per bout between training day i and the baseline day in the convenient unit of SDs.

Zebra finch song is mostly linear, with occasional rare repetitions within a motif. However, individual birds can stereotypically repeat introductory elements and ending elements of a motif 1–3 times. We focused analysis on vocal elements or song syllables that were observed to repeat two or more times on at least one occasion during optical manipulations or following knockdown of FoxP2 in Area X. Once identified, we retrospectively tracked these syllables back to baseline periods or forward in time to recovery periods in order to examine whether repetitions changed during our manipulations.

*Quantification of changes in syntax*. Syntax alterations are defined as new syllable transition(s) (i.e., transitions not present prior to FoxP2 knockdown) occurring in a song motif. Syllables omissions are dropped syllables that were present in the song motif prior to FoxP2 knockdown. Syllable additions are as insertion of de novo created syllable(s) (i.e., syllables not present prior to FoxP2 knockdown) into the song motif. Omission and addition of syllables at the beginning or end of a song motif are not classified as syntax alterations. Quantification of percent syllable alteration, omission, and addition was calculated from 30 song bouts at each timepoint presented in the figures. Changes in song sequences refers to any change in a song bout which alters either the syntax, syllable transition probability or frequency of vocal repetitions. For example, in the song in Fig. 2e syllables 'ab' and 'ghj' are consistently dropped after FoxP2 knockdown. Syllables 'ab' and 'ghj' were noted to also be dropped in a subset of motifs at baseline. The consistent dropping of these syllables is considered a change in song sequence. These and other changes in the song are captured by changes in the syllable transition matrix (Fig. 2e) and syllable repetitions but are not considered syntax alterations.

*Syllable transition matrix*. Song files (29 ± 1 (mean ± SEM)) reflecting a total of 140–458 s (299.7 ± 26.9 (mean ± SEM)) of song per bird per timepoint (baseline, indicated months following CS-shFoxP2 or Cre-GFP injection), were used for the analysis of syllable transition probabilities. Syllables in those song files were labeled by hand by an expert based on their song spectrograms using a custom MATLAB program. Syllable labels, onset times, and offset times were then exported to R for further analysis. Silent gaps of over 100 ms between the offset of one syllable and the onset of the following syllable were considered song boundaries and are denoted by the character "/". Syllables which are preceded by and followed by a 100 ms silence are removed as they likely reflect calls rather than song syllables. Each pairwise transition between syllables or to a song boundary is tallied to determine the probability of each of the possible syllables occurring following a given current syllable. Syllables are described as "variants" of each other if they appear spectrally similar, but slight modifications to a common motif syllable are observed in different syntactic contexts. Syllable variants (for example f' and f in Supplementary Fig. 2) were treated as separate syllable types in this analysis and were determined by consensus of two experts when labeling syllables based on their spectrograms. Syllables which were present in fewer than 20% of all song bouts at any timepoint were omitted from the difference matrices in Figs. 2d, 2e, 3e, Supplementary Figs. 2b, 2c, 2e for clarity.

*Song similarity and variability*. SAP2011 was used to quantify the song similarity and variability between different conditions in reversible FoxP2 knockdown experiments[92]. To measure song similarity across conditions, motifs from 30 song bouts on a given day (introductory element excluded, recorded ~4–6 h after lights on) were compared to a single representative motif from baseline day to generate similarity scores (symmetric comparisons). We used the average %similarity measure and %sequential similarity measures respectively for our %acoustic similarity to baseline and %sequential similarity to baseline analysis. For song similarity-to-self measurements, the same set of motifs used above was compared to a single representative motif from the same condition to generate self-similarity scores (symmetric comparisons). We used the CV of self-similarity scores from these comparisons as a measure of how the variability of self-similarity changed in the same bird across different conditions. To measure the change in vocal variability between baseline and post-knockdown timepoints, all syllables with constant frequency components (i.e., harmonic stacks) from 30 motifs for each condition had their pitch (YIN method) and Weiner entropy measured. We used the CV of pitch and Weiner entropy to quantify the change in variability of acoustic features following FoxP2 knockdown. Because the syntax was altered in a subset of birds following reversal knockdown of

FoxP2 relative to baseline, comparisons between two different conditions were restricted to the preserved vocal elements. For birds who developed more than one representative motif following reversal of FoxP2 knockdown, we pooled all representative motifs and used a single one for individual comparisons.

**Immunohistochemistry and immunoblotting**. Birds were anesthetized with Euthasol (Virbac, TX, USA) and transcardially perfused with ice-cold phosphate-buffered saline (PBS), followed by 4% paraformaldehyde (PFA) in PBS. The brains were post-fixed in 4% PFA for 2 h at 4 °C, then transferred to PBS containing 0.05% sodium azide. The brains were sectioned at 50 µm, using a Leica VT1000S vibratome. Immunohistochemistry (IHC) was performed as described previously[38]. Sections were first washed in PBS, then blocked in 10% normal donkey serum (NDS) in PBST (0.3% Triton X-100 in PBS) for 1 h at room temperature (RT). Sections were incubated with primary antibodies in blocking solution (PBST with 2% NDS and 0.05% sodium azide) for 24–48 h at 4 °C, then washed in PBS before a secondary antibody incubation (1:500, anti-chicken Alexa Fluor 488,703-545-155; anti-rabbit DyLight 405,711-475-152; anti-goat Alexa Fluor 647,705-605-003; anti-mouse Alexa Fluor 594,711-585-150 or anti-rabbit Alexa Fluor 594,711-585-152, Jackson Immuno Research, ME, USA) for 4–6 h at RT. Images were acquired with an LSM 710 laser-scanning confocal microscope (Carl Zeiss, Germany), processed in Zen Black 2012 and analyzed in ImageJ(1.52p). To minimize bias during quantification, matched regions across animals were selected for IHC according to GFP expression levels. To mitigate any crosstalk between distinct fluorescent channels, spectral unmixing was used and a region of interest (ROI) around each mCherry+ or tagBFP+ cell was manual drawn and overlaid on the channel (Alexa Fluor 647) designed for FoxP2. The expression level of FoxP2 in each ROI was estimated by measuring the mean intensity value from the nucleus of each ROI and subtracting the background intensity. The expression levels of FoxP2 in control cells were estimated from 20 randomly selected non-infected cells (FoxP2+mCherry−tagBFP−) from the same slice where the expression in mCherry+ and tagBFP+ cells were measured.

Immunoblotting (IB) was carried out as described previously[99]. Cell lysates from each sample were separated by SDS-PAGE and transferred to an Immuno-Blot PVDF Membrane (162-0177, Bio-Rad Lab., CA, USA), then blocked with 1% skim milk in TBST (tris-buffered saline with 0.1% Tween-20) for 1 h at RT. The membrane was incubated with primary antibodies overnight at 4 °C, washed in TBST, and reacted with the appropriate horseradish peroxidase (HRP)-conjugated species-specific secondary antibodies (1:10,000, anti-rabbit NA934 and anti-mouse NA931, Sigma-Aldrich, MO, USA; 1:10,000, anti-goat AP180P, Millipore, MA, USA) for 1 h at RT. The signals were detected by Clarity western ECL substrate (170-5060, Bio-Rad Lab., CA, USA).

The primary antibodies used were: Goat anti-FOXP2 (1:500, ab1307, Abcam, MA, USA), Goat anti-FOXP2 (1:500, sc-21069, Santa Cruz Bio., TX, USA), mouse anti-V5 tag (1:2000, R960-25, Invitrogen, CA, USA), rabbit anti-RFP (1:1000, mCherry, 600-401-379, Rockland, PA, USA), mouse anti-RFP (1:1000, mCherry, 200-301-379, Rockland, PA, USA), rabbit anti-GFP (1:2000, A11122, Invitrogen, CA, USA), chicken anti-GFP (1:1000, AB16901, Millipore, MA, USA), rabbit anti-tRFP (1:200, tagBFP, AB233, Evrogen, Moscow, Russia), mouse anti-GAPDH (1:20,000, MAB374, Millipore, MA, USA) and rabbit anti-beta Tubulin(1:20000, ab6046, Abcam MA, USA). The specificity of primary antibodies against FoxP2, mCherry or GFP were confirmed by two independent primary antibodies for both IHC and IB.

**Statistics**. The Shapiro–Wilk test was performed for all behavioral data to test for normality of underlying distributions. Unless otherwise noted, non-parametric two-tailed statistical tests were performed; Wilcoxon signed-rank tests and Mann–Whitney tests were used where appropriate. Kruskal–Wallis tests were performed when comparisons were made across more than two conditions (e.g., baseline vs stimulation day vs recovery day) from individual animals, whereas Friedman tests were performed when data was pooled across animals and comparisons were made across more than two conditions. Statistical significance is represented as *$p < 0.05$, **$p < 0.02$. Statistical analysis was performed using Prism 6.0 (GraphPad Software, USA). Statistical details for all experiments are included in their corresponding figure legends.

**Tissue processing for snRNA-seq**. Adult zebra finches (120–140 dph) were injected with CS-shFoxP2 ($n = 2$) or CS-shScr ($n = 2$) constructs and sacrificed at 180–200 dph. Birds were put down prior to lights-on to ensure that recent singing behavior did not affect our results. Each bird was rapidly decapitated, and its brain was placed in ice-cold ACSF (126 mM NaCl, 3 mM KCl, 1.25 mM NaH$_2$PO$_4$, 26 mM NaHCO$_3$, 10 mM D-(+)-glucose, 2 mM MgSO$_4$, 2 mM CaCl$_2$) bubbled with carbogen gas (95% O$_2$, 5% CO$_2$). The cerebellum was removed with a razor blade and the cerebrum was glued to a specimen tube for sectioning with a VF-200 Compresstome (Precisionary Instruments). Coronal 500 µm sections were made in ice-cold ACSF and allowed to recover in room temperature ACSF for 5 min. Area X punches were placed into a tube containing ACSF on ice until all punches were collected and pooled from 2 birds per condition. Tissue punches were dounce-homogenized in 500 µl ice-cold Lysis Buffer (10 mM Tris pH 7.4, 10 mM NaCl, 3 mM MgCl$_2$, 0.1% IGEPAL CA-630) and transferred to a clean 2 ml tube. Then, 900 µl of 1.8 M Sucrose Cushion Solution (NUC201-1KT, Sigma, MO, USA) was added and pipette-mixed with nuclei 10 times. 500 µl of 1.8 M Sucrose Cushion

## Table 1 Gene markers for cluster identities.

| Gene symbol | Gene name | Identity | Reference |
|---|---|---|---|
| Gad2 | Glutamate decarboxylase 2 | GABAergic | 44 |
| Slc17a6 | Solute carrier family 17 member 6 | Glutamatergic | 44 |
| Lhx6 | LIM homeobox 6 | MGE-derived | 101 |
| Pvalb | Parvalbumin | Interneuron (ZF) | 43 |
| Sst | Somatostatin | Interneuron (ZF) | 43 |
| Npy | Neuropeptide Y | Interneuron (ZF) | 43 |
| Nos1 | Nitric oxide synthase 1 | Interneuron (ZF) | 43 |
| Chat | Choline O-acetyltransferase | Interneuron (ZF) | 43,50 |
| Penk | Proenkephalin | PN (ZF) | 50 |
| Tshz1 | Teashirt zinc finger homeobox 1 | PN (ZF) | Present study |
| Lrig1 | Leucine rich repeats and immunoglobulin like domains 1 | Astrocyte | 102 |
| Meis2 | Meis homeobox 2 | LGE-derived | 101 |
| Ppp1r1b | Protein phosphatase 1 regulatory subunit 1B (also known as DARPP-32) | MSN (ZF) | 43,44 |
| FoxP1 | Forkhead box P1 | MSN (ZF) | 45 |
| FoxP2 | Forkhead box P2 | MSN (ZF) | 17,45 |
| Tac1 | Tachykinin precursor 1 | MSN (ZF) | 43,50 |
| Csf1r | Colony stimulating factor 1 receptor | Microglia | 102 |
| Flt1 | Vascular endothelial growth factor receptor 1 | Endothelial | 102 |
| Mbp | Myelin basic protein | Oligodendrocyte | 102 |
| Pdgfra | Platelet-derived growth factor receptor A | Oligodendrocyte Precursor | 102 |

Functional cluster identities were assigned based on established gene markers from previous studies. When the study was specific for zebra finches, this is noted with "(ZF)".

Solution was added to a second clean 2 ml tube, and the nuclei sample was layered on top of the cushion without mixing. The sample was centrifuged at $13,000 \times g$ for 45 min at 4 °C and all but ~100 μl of supernatant was discarded to preserve the pellet. The pellet was washed in 300 μl Nuclei Suspension Buffer (NSB) (1% UltraPure BSA (AM2618, Thermo Fisher Scientific, MA, USA) and 0.2% RNase inhibitors in PBS) and centrifuged at $550 \times g$ for 5 min at 4 °C. All but ~50 μl of supernatant was discarded and the pellet was resuspended in the remaining liquid and filtered through a FLOWMI 40 μm tip strainer (H13680-0040, Bel-Art, NJ, USA). Samples were diluted to 1000 nuclei/μl with NSB to yield 10,000 nuclei for snRNA-seq. Libraries were prepared using the Chromium Single Cell 3′ Library & Gel Bead Kit v3 according to the manufacturer's instructions and sequenced using an Illumina NovaSeq 6000 at the North Texas Genome Center at UT Arlington.

**Pre-processing of snRNA-seq Data.** Raw sequencing data were obtained as binary base cells (BCL files) from the sequencing core. 10X Genomics CellRanger v.3.0.2 was used to demultiplex the BCL files using the *mkfastq* command. Extracted FASTQ files were quality checked using FASTQC v0.11.5. Paired-end FASTQ files (26 bp long R1 —cell barcode and UMI sequence; 124 bp long R2—transcript sequence) were then aligned to a reference zebra finch genome (bTaeGut1_v1.p)[100] from the UCSC Genome Browser, and reads were counted as the number of unique molecular identifiers (UMIs) per gene per cell using the 10X Genomics CellRanger v.3.0.2 *count* command. Since the libraries generated are single-nuclei libraries, reported UMIs per gene per cell account for reads aligned to both exons and introns. This was achieved by creating a reference genome and annotation index for pre-mRNAs.

**snRNA-seq clustering analysis.** The resulting count matrices from the data pre-processing steps were analyzed with the Seurat analysis pipeline in R (v.3.0, https://satijalab.org/seurat/v3.0/pbmc3k_tutorial.html). Cells with more than 10,000 UMI and more than 5% mitochondrial genes were filtered out to exclude potential doublets and dead or degraded cells (Fig. S8c–f). As described in the Seurat pipeline, the data were log-normalized, and scaled by a factor of 10,000, and regressed to the covariates of the number of UMI and percent mitochondrial genes. Top variable genes were identified, and principal components (PCs) were calculated from the data. PCs to include were identified by "ElbowPlot" in Seurat, where PCs are ranked according to the percentage of variance each one explains; PCs were excluded after the last noticeable drop in explanatory power. With the selected PCs, the Louvain algorithm was then used to identify clusters within the data. Clusters were visualized with uniform manifold approximation and projection (UMAP) in two dimensions. For hierarchical clustering, differentially expressed genes across the dataset were identified with the Wilcoxon Rank Sum test (Seurat FindAllClusters, min.pct = 0.1, logfc.threshold = 0.25, max.cells.per.ident = 200), the normalized data was averaged within each cluster, and the 50 genes with the top fold-change for each cluster were then used for hierarchical clustering (function *pvclust* from the R package *pvclust v2.2-0*, correlation distance, average agglomeration, 100 bootstrap replicates).

**snRNA-seq data integration and differential gene expression.** In order to compare gene expression between CS-shScr+ and CS-shFoxP2+ birds, the datasets must first be combined. Each dataset was processed independently as described in

Clustering Analysis up until the point of normalization. Once normalized, the datasets were integrated as described in the Seurat integrate pipeline. The integrated dataset was then regressed to the covariates of the number of UMI and percent mitochondrial genes. The subsequent analysis proceeded as described in Clustering Analysis. The integrated data were used to generate the integrated clusters and UMAP plot, as recommended by the Seurat pipeline (https://satijalab.org/seurat/v3.0/immune_alignment.html). Comparisons of gene expression across clusters and across datasets were pulled from each individual RNA assay, as recommended by the Seurat pipeline. Differentially expressed genes were identified using the Wilcoxon ranked sum test with the Bonferroni correction as implemented in the Seurat pipeline. For gene ontology analysis, the top differentially expressed genes in FoxP2+ cells between CS-shScr+ and CS-shFoxP2+ birds were loaded into ToppGene (https://toppgene.cchmc.org) with default settings (false discovery rate correction applied; *p*-value cutoff 0.05; gene limits $1 \leq n \leq 2000$; random sampling size 5000; minimum feature count 2) and then gene family categories were filtered for similarity with REVIGO (http://revigo.irb.hr) on the "small (0.5)" similarity setting. For an analysis of differentially expressed genes related to autism in FoxP2+ cells of CS-shFoxP2+ birds, all genes were aligned with all scored genes in the Simons Foundation Autism Research Initiative (https://www.sfari.org), where Category 1 is "high confidence."

**Cluster function assignment.** Clusters were assigned functional identities based on the expression of gene markers established in the literature (see Table 1). Identity names were as specific as possible given the confidence and clarity of the gene expression pattern in relation to expression in other clusters. When gene markers have been established in Area X of zebra finches specifically in addition to mammals, this is noted in the table with a "ZF". For downstream analyses, specific UMAP coordinates were used to further specify MSNs (UMAP_1 > −5 & UMAP_2 > −6) and PNs (UMAP_1 > 0 & UMAP_2 > −7.5).

**Reporting summary.** Further information on research design is available in the Nature Research Reporting Summary linked to this article.

## Data availability

The NCBI Gene Expression Omnibus (GEO) accession number for the raw snRNA-sequencing data in this manuscript (Figs. 4–6) is GSE136086. Processed data for snRNA-sequencing analyses are available at https://cloud.biohpc.swmed.edu/index.php/s/nLicEtkmjGGmRF8. Source data for Figs. 1–3 and 7–8 are provided with the paper, all other data associated with this paper are available from the corresponding author upon reasonable request.

## Code availability

Code for snRNA-sequencing data, including pre-processing, clustering, differential gene expression analyses, and producing all related figures are available on GitHub (https://github.com/konopkalab/songbird_areax). Other source code for closed-loop manipulations and analysis of behavioral data associated with this paper are available from the corresponding author upon reasonable request.

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

## Acknowledgements

The authors thank members of the Roberts and Konopka laboratories for discussion and comments on the manuscript, Jennifer Holdway and Matthew Harper for laboratory support, Gaurav Chattree for helping with optogenetic experiments, Maaya Ikeda and Chung Yan Cheung for advice on reverse microdialysis experiments, Clarissa Fuentes for behavioral analysis, Andrea Guerrero for animal husbandry and song recording, Ashley Anderson for cloning of FoxP2 V5, and Erich Jarvis for the original clone of zebra finch FoxP2. This research was supported by grants from the US National Institutes of Health R21DC016340 to T.F.R. and G.K., R01NS102488 to T.F.R. and R01DC014702 to G.K. DPM was supported by F32NS112557.

## Author contributions

L.X., D.P.M., and T.F.R. designed the experiments and wrote the manuscript, L.X. collected and analyzed the optogenetic, pharmacological, and gene knockdown experiments, and helped collect the snRNA-seq data, DPM analyzed the snRNA-seq data, T.M.I.K. developed analysis tools and helped analyze behavioral data, Mou Cao analyzed and imaged the anatomical data and Marissa Co collected the snRNAseq data, A.K. developed bioinformatic pipelines for snRNA-seq data analysis and helped analyze the data, G.K. supervised the snRNA-seq data collection and analysis and helped design the reversible gene knockdown experiments, T.F.R. supervised all experiments. All authors read and commented on the manuscript.

## Competing interests

The authors declare no competing interests.
