## [Peer Review File · Nature Communications]

Reviewers' Comments:

Reviewer #1:

Remarks to the Author:

Review of Xiao et al. 2020, Nature Communications

Building on prior work, the authors study the relationship between knockdown of the speech gene FoxP2 in finch basal ganglia song nucleus Area X and dopamine signaling on behavioral output. They use an impressive multi-faceted methodological approach from their construct design to the use of single cell sequencing and optogenetics to produce a substantial body of experimental work that sheds important insight on the functional role of FoxP2 and dopamine signaling in vocal production.

Manuscript revision is needed to address concerns as detailed below.

Introduction:

This section is brief and requires a more detailed presentation of the background literature and clarification of concepts such as 'vocal variability' for a non-songbird audience. It is clear that their methodological approach is novel but the authors need to clarify how their work breaks new ground conceptually in comparison to the previous literature in this area. Specific questions/suggestions are as follows:

1) Line 28: provide a citation for FoxP2 expression in medium spiny neurons

2) Line 32: Multiple reviews are cited instead of the primary literature that has used viral-mediated FoxP2 manipulation in Area X to impact vocal learning, so I recommend citing those instead and one review -e.g. Haesler et al. 2007 JNeurosci, Murugan et al. 2013 Neuron, Heston and White, 2015 JNeurosci, Burkett et al. 2018 eLife. Then, the next paragraphs should provide the salient details from these studies in a logical manner.

3) Lines 34-43: This section needs to be greatly expanded to lay out the case for FoxP2 in:
-driving phonology versus syntax during learning and adult song maintenance taking into account the papers above
-juvenile song learning: imitation/copying of tutor's song
-clarify the concept of vocal variability for non-songbird researchers and how it is measured and generated by the cortico-basal ganglia-thalamo cortico loop.

4) Lines 44-51: some of these references are from mice not songbirds making it confusing as to which study showed what when the case is being made for finch Area X.

5) Dopamine ablation: Line 51: "whereas ablating dopaminergic input in adult birds has not been reported to disrupt the ability to accurately produce or maintain song." This statement is not correct and depends upon the experimental methods used- see Miller, Hafzalla et al. 2015, Physiological Reports - Injection of 6-hydroxydopamine into zebra finch Area X reduces vocal variability in undirected adult birdsong (accuracy scores go up). Miller et al. were not testing this neurotoxin in an auditory disruptive learning task as per Hoffman et al. 2016 so the methodology was different and the species (zebra finch vs Bengalese where zebras have less complicated syntax). Also see Hara et al. 2007, European Journal of Neuroscience, where 6-OHDA unilateral injection into VTA/SN decreased singing rate in female-directed but not undirected song. It should also be noted that manipulation of dopamine input from VTA/SN to Area X can impact song/social context - see papers by Gadagkar Science 2016, et and also LeBlois and Perkel 2010 J Neurosci where dopamine receptor blockers affect vocal variability in adult birdsong.

Given the focus on dopamine signaling in this paper, I recommend a more careful review of the literature about the role of dopamine in Area X and how the authors' approach provides new information. It can be summarized efficiently if there is a length restriction.

6) Lines 52-57: The topic jumps in this paragraph from Huntington's disease to FoxP2 manipulation in deafening animals. It would be helpful to have a discussion about the findings of the Murugan 2013 study in which FoxP2 knockdown in Area X has effects on song and social context and dopamine signaling (as per comment #5 above). Then, lead into the novelties of your study and methodology. The power of your study is in the experimental design and execution-the first evidence for FoxP2 rescue in songbirds and a direct role for dopamine signaling/interaction with FoxP2. If it is the first evidence for FoxP2's role in sequencing, that should also be stated.

Results:

1) Line 31: Define the term song linearity as this is its first use (=is it vocal repetition?) vs syntax.

2) In addition to measuring repetitions and syntax, the authors examined whether vocal variability was altered by FoxP2 knockdown in Area X by measuring the CV of pitch and entropy. Did they examine phonological measures that might change in song structure such as similarity scores (see point # 2 in Discussion)? How many song renditions and syllables per bird over what length of time were used for the calculations? this information was absent from the Methods.

3) Fig. 2b: given the short and similar motifs across the spectrograms, it would be helpful to use letters or numbers underneath the spectrogram to denote the syntax and make it easier to see the switch in syllable order, etc. A Markov chain figure would also help to show the difference in syllable transitions between the control and knockdown groups.

4) Fig. 2d: define motif versus bout vs syllable for the reader in the spectrogram legends so that the naïve reader does not have to go back and forth to the Methods text.

5) Fig. 2d-e: indicate the time point (one, two or four months) post-virus injection here and whether the other timepoints were not significant between controls and shFoxP2.

6) Fig. 5d-more clarification on the scale below the x-axis is needed.

7) Fig. 6-explain the left vs right shift of the box plots-is it meaningful or is just the height of the plots meaningful? This information is difficult to parse-can you state more clearly what the top vs bottom trace shows in each panel?

8) Line 338-were dopamine infusions done in undirected finches only-clarify. Also, since you did not find changes in spectral or temporal structure with dopamine infusion/blockers, this is worth comparing to the LeBlois and Perkel 2010, 2012 papers that did find changes in vocal variability particularly with D2 receptors.

Discussion:

The Discussion is an excellent and thoughtful summary of the results with respect to how FoxP2-dopamine pathways interact and with relevance to human speech disorders.

Specific comments below-

1) Line 447-Restoring FoxP2 expression later in adulthood resulted in recovery of linear song syntax. This sentence conveys that your rescue phenotype improved syntax yet in the Results and figures, you stated that it only reduced vocal repetitions and that there were still weird syntax changes (including insertion of novel elements).

2) Line 482-However, we did not observe spectral degradation of individual song syllables, as seen following deafening. In order to be confident in this statement, it would be necessary to measure other spectral features of

song in knockdown vs control animals besides CV of entropy and pitch presented here. For example, Haesler et al. 2007 and Heston et al. looked at virus-mediated changes in similarity (accuracy) scores in juveniles as well as CV of multiple acoustic features (duration, amplitude, FM). Can you do select further analysis to confirm?

3) Line 534: extra 'to' in this sentence... birdsong and speech plans into to fluent vocal-motor actions.

4) It is worth commenting in this discussion on why you did not observe effects of dopamine infusion and receptor blockade on song features given evidence from the LeBlois and Perkel work. What does this mean for the Sasaki et al. 2006 paper which showed that dopamine levels are higher in Area X for FD but not UD?

Methods:

1) Line 698-Typos/extra words in this sentence: scarified at 180-200dph. Birds were put down prior to lights in the to ensure that singing behavior did not

Supplements:

1) Fig. S3-clarify the presence of the second black line in panel b and e.

Reviewer #2:

Remarks to the Author:

This paper has 3 main components; (1) effect of FOXP2 knockdown in adult area X on singing behaviour, (2) single nuclei analyses of area x subtypes and consequences of knockdown on dopamine expression and (3) effect of optogenetic stimulation of dopamine circuits on singing behaviour. The individual studies are beautifully executed, however the 3 do not quite come together as a whole. With some revisions and new data analyses it is likely that they could come together more coherently. The main concerns regarding the study are:

1. The single nuclei sequencing study seems far more focused on establishing cell types to say something about the circuitry of area X than looking at the role of FOXP2 or consequences of its disruption. The point of the section on lines 226-286 seems to be trying to prove something about direct- and indirect-like pathways/populations in area X, which is not well supported, nor does it fit with the narrative of the paper. It would be better instead to use this excellent data to draw conclusions about the role of FOXP2 in these cell populations and establish the identity of the populations so that any disruptions to expression patterns could be explored thoroughly after FOXP2 knockdown (see also 2). The paper should also be very careful about drawing parallels with the mammalian direct/indirect pathway system when only relying on sequencing data and without neurophysiological data to support the conclusion that "MSNs in Area X can segregate into broad classes corresponding to direct-like (Drd1+/FoxP2+) and indirect-like pathways (Drd2+/FoxP2-) (Figure 5a), as well as a third Drd1/2+ class.". In fact, just from the sequencing data, I do not see the parallels to direct/indirect categories since that FOXP2 positive cells are prominent in every group, including the DRD2 positive groups in Figure 5.

2. The single cell sequencing data is sorely under used to determine consequences of FOXP2 knockdown – only consequences on dopamine receptors are looked at, whereas there is an enormous and valuable dataset that is basically ignored. This data set should be fully exploited to determine the molecular consequences that result from FOXP2 knockdown. Such a fine-grained molecular analysis of FOXP2 disruption has never been reported in birds and would represent an important resource for the field and may give important new insight into the function of FOXP2, rather than limiting the study to only confirm previous findings that FOXP2 regulates dopamine receptors. This is not to malign Figure 6 - it is very well done, but it is unfortunate not to go further with other genes given the dataset at the authors disposal.

3. The link between the optogenetic studies and FOXP2 are tenuous. It is not clear that these optogenetic studies tell us anything about FOXP2 circuitry. As it stands the experimental links between the FOXP2 knockdown and consequences of phasic dopamine are somewhat disjointed and coincidental in their relationship, rather than being directly proven. From the text it is not entirely clear to me what can be really taken away from this combination of experiments. If the optogenetic studies were to be combined with the FOXP2 knockdown approaches in the same animal there could be some very exciting proof of function experiments reported.

Minor comment – many references in the paper seem to reference review papers rather than the original reporting of findings. It would be much better to where possible refer to the original research than being overly reliant on review papers.

I would like to highlight that the data analyses and figures are really excellently done – the data representation makes the findings very clear and are beautifully represented. Also, the experimental design of the inducible Cre FOXP2 knockdown system is superb.

Reviewer #3:

Remarks to the Author:

General comments:

This study by Xiao et al on the role of FoxP2 in singing behavior, physiology, and gene regulation in adult songbirds puts lots of pieces of a puzzle together, to come out with exciting findings. With an apparently stronger knock down of FoxP2 in the striatal song nucleus Area X, they found very little acoustic changes, but syntax and stuttering deficits, more like what is seen with humans with a FoxP2 mutation. They are able to recover (repair) the deficits using a Cre reversal vector. They, for the first time, perform single nuclei transcriptome sequencing within the songbird striatum, characterize the cells types more robustly, and make needed advances in relationship with mammals. They find that the knockdown changes the balance in the D1 and D2 receptor families, in manner consistent with the behavioral findings and physiology of the striatum. This leads to a hypothesis of dopaminergic function, which they test with optogenetic stimulation of dopaminergic synthesizing neurons, which causes some similar behavioral deficits as the FoxP2 knockdown. This paper has it all, is multidisciplinary, and makes important advances in our understanding of the role FoxP2 plays in vocal learning circuits, song and speech. I am very excited by the study.

I do have some concerns, which I think addressing them will improve the manuscript. The authors were pretty good as relating their findings to the past literature, but missed out on a several relevant important studies.

The authors do not relate their findings to the FoxP2 studies and vocal behavior in rodents. The most relevant are the Castelluci et al 2016 (Scientific Reports) and Chabout et al 2016 (Frontiers) studies. They found that the heterozygous FoxP2 mutant mice with the same or similar mutation as humans had very little to no acoustic changes in their syllables, but had sequencing deficits. The males did not readily switch to the more complex sequences that they make in the presence of females, and instead continued to produce the simpler syllable types, with reduced syntax variability. They suggested that these sequence deficits are similar to the deficits seen in humans, where the underlying problem is difficult for us humans to sequence a complex series of phonemes into complex words. Chabout et al argued that this finding in mice and humans was different than the Heassler et al 2007 study in zebra finches, where the later found mostly acoustic deficits in learning but little to no sequencing deficits. They hypothesized that knocking down FoxP2 in the striatum only (in songbirds) may have a different deficit than heterozygous partial knockdown in the whole brain (humans and mice). Xiao et al now find that they can also get sequencing deficits in adult songbirds with a local striatal knockdown, refuting this explanation for differences between studies. What the combined studies suggest to me is that like Chabout et al stated, FoxP2 may have been utilized for sequential (and possibly acoustic) control of vocalizations by the striatum, before vocal learning evolved; but that when the vocal learning circuitry

appeared or became highly specialized in songbirds and humans, the circuit became even more dependent on the gene. I think the authors need to consider these rodent findings and the associated hypotheses in the interpretation of their findings as well.

As the authors noted, some of the stuttering findings after adult Area X lesions in the Kubikova et al 2014 (Scientific Reports) is similar to the FoxP2 Area X knockdown stuttering findings in the current study. But, I don't think the authors go far enough in comparing and contrasting these findings. Kubikova et al found stuttering mainly on the motif end syllables, if the bird had a predisposition of some repeats before the lesions, and only with chemical lesions that left fibers intact, and not electrolytic lesions. They further found a relationship between stuttering when new neurons enter Area X to repair the nucleus. They did not find stuttering with a chemical LMAN lesion; but Chakraborty et al 2017 (Scientific Reports) later found stuttering with viral NR2B knockdown in LMAN, both at the beginning and ends of song motifs. All this is to say that when combined with the findings of Xiao et al, it looks like some manipulations and conditions lead to stuttering, to different types of stuttering, and others do not, including when using different viral vectors to knock down FoxP2. I do believe the author's interpretation that Area X is involved in the initiation and stopping of motifs, and syntax is plausible. But I think multiple things have to line up for this to happen. BTW, the authors should also credit Kubikova et al 2014 for an initial proposal for such an idea "we propose that LArea X may be involved in providing cues for the initiation and selection of the next motor segment in a song sequence in adult animals, and the selection is more critical in individuals with learned repetitions at the end of their motifs".

It was good to see that the authors measured both D1 and D5 dopamine receptors in the striatal neurons, where many studies only look at D1. I think they should consider the alternative names for these two receptors, D1A and D1B respectively, whose evolution and a revised terminology was mapped out in two other studies Kubikova et al JCN (2010) and Yamamoto et al 2013 (Mole Eco Evol). Both receptors belong to the D1 family, whereas the D2 family has a different evolutionary origin, converging on binding to dopamine. Although the authors did measure that three dopamine receptors that are found in humans (D1, D5, and D2), birds also have three additional receptors (D1C, D3 and D4), with the former two also expressed in the brain, although not the striatum (Kubikova et al 2010). D3 and D4 belong to the D2 family. I think it will be worthwhile mentioning this to the readers.

How do the authors interpret their findings in the context of those from the White lab (Teramitsu et al 2004 J. Neurosci) that singing causes down regulation of FoxP2 expression in Area X? Considering that they are doing a knockdown.

Specific comments:

Line 34. State that Area X is a song nucleus within the songbird striatum.

Figure 1. Panel d: What are the individual values? Where is the triple label image?

Line 193. It is good that the authors describe the variability of findings from individual animals. They should keep such descriptions.

Line 215. Here the authors used a paired t-test, and elsewhere use Wilcoxon signed rank or other test when comparing the same animals at two different time points. The authors need to be consistent.

Line 227. I suggest changing from FoxP2 is expressed in "Area X" to "Area X and the surrounding striatum".

Line 240. Arkypallidal cells. The author should cite a paper for this cell type (have not heard of it before), and perhaps include a panel that shows the result of the cell type also in Area X.

Line 261. No citation is given to these findings of D1 and D2 overlap in the prior study. Do they mean

to cite Kubikova et al 2010?

Line 458. But did the prior mammal studies also include D5 (D1B) in their studies? Most have not.

Line 474. What is the difference between sequences and syntax? I see in the methods, maybe the authors mean changes at the beginning and end of a motif for sequences, and the middle for syntax? This is not a common, nor I think a good, definition. I would call all of this syntax or sequence changes, and their locations defined as end or internal syntax changes.

Line 533. In light of the mouse findings, convergence of FoxP2's role in vocal sequencing could have been inherited in songbirds and humans, and then enhanced in the vocal learners independently.

Line 545. It would be helpful to add a phrase in some of the sentences to indicate what each virus was used for, like FoxP2 overexpression, knockdown, etc. Otherwise the sentences read as if they are one continuous protocol for just one virus construct.

Line 600. Did the authors select good singers before surgeries, and if so, how? I imagine it will be hard to get animals to sing after many of these manipulations.

Line 613. Where the songs for analyses selected manually or automatically?

Line 639. I am not following this paragraph well, but it seems like the authors are filtering out songs without repeats before analyses? That would be a bias?

Line 704. Where the Area X punches made under a microscope? Was it possible to avoid the surrounding striatum? What is the degree of spread of the virus inside and outside of Area X? Did the authors use fluorescent markers to identify the injection site for the punch?

Line 725. Cite Rhie et al 2020 biorxiv for the new zebra finch assembly used.

Figure S1. Looks like the injections respect the boundary of Area X? This would be strange. Area X is also not perfectly round as the dashed lines in panel e indicate. Which direction is front of the brain?

Figure S2. I am surprised that the last value at month 5 is not significant in the first graph? Why are errors bars on some months but not others?

Figure S4. What does the yellow circle mean in panel c? Shouldn't this panel come at the end of the figure?

Figure S7. The green circle in panel c is not the shape of Area X.

We thank the reviewers for their thoughtful comments and suggestions. To fully respond, we have undertaken a deeper analysis of our behavioral and snRNA sequencing data, adding several new figure panels in the main and supplemental figures and 4 completely new supplemental figures. In addition, we have extensively edited the Introduction, Results, and Discussion sections in accordance with the reviewer's suggestions. We provide a revised manuscript with track changes. Below we provide detailed responses to each of the reviewers' comments. Call-outs for line numbers in the manuscript reflect those when track changes are hidden.

Reviewer #1 (Remarks to the Author):

Review of Xiao et al. 2020, Nature Communications

Building on prior work, the authors study the relationship between knockdown of the speech gene FoxP2 in finch basal ganglia song nucleus Area X and dopamine signaling on behavioral output. They use an impressive multi-faceted methodological approach from their construct design to the use of single cell sequencing and optogenetics to produce a substantial body of experimental work that sheds important insight on the functional role of FoxP2 and dopamine signaling in vocal production. Manuscript revision is needed to address concerns as detailed below.

Introduction:

This section is brief and requires a more detailed presentation of the background literature and clarification of concepts such as 'vocal variability' for a non-songbird audience. It is clear that their methodological approach is novel but the authors need to clarify how their work breaks new ground conceptually in comparison to the previous literature in this area. Specific questions/suggestions are as follows:

1) Line 28: provide a citation for FoxP2 expression in medium spiny neurons

We now include a citation for this.

2) Line 32: Multiple reviews are cited instead of the primary literature that has used viral-mediated FoxP2 manipulation in Area X to impact vocal learning, so I recommend citing those instead and one review -e.g. Haesler et al. 2007 JNeurosci, Murugan et al. 2013 Neuron, Heston and White, 2015 JNeurosci, Burkett et al. 2018 eLife. Then, the next paragraphs should provide the salient details from these studies in a logical manner.

We have replaced multiple review papers with original studies which were cited elsewhere and provided more salient details from them in the following paragraph.

3) Lines 34-43: This section needs to be greatly expanded to lay out the case for FoxP2 in:

- driving phonology versus syntax during learning and adult song maintenance taking into account the papers above
- juvenile song learning: imitation/copying of tutor's song
- clarify the concept of vocal variability for non-songbird researchers and how it is measured and generated by the cortico-basal ganglia-thalamo cortico loop.

We have expanded our introduction in accordance with these constructive suggestions. Please see lines 34-52 of our submission.

4) Lines 44-51: some of these references are from mice not songbirds making it confusing as to which study showed what when the case is being made for finch Area X.

We apologize for those confusing references. We have distinguished between work from rodents and songbirds more explicitly in the revised manuscript.

5) Dopamine ablation: Line 51: “whereas ablating dopaminergic input in adult birds has not been reported to disrupt the ability to accurately produce or maintain song.” This statement is not correct and depends upon the experimental methods used- see Miller, Hafzalla et al. 2015, Physiological Reports - Injection of 6-hydroxydopamine into zebra finch Area X reduces vocal variability in undirected adult birdsong (accuracy scores go up). Miller et al. were not testing this neurotoxin in an auditory disruptive learning task as per Hoffman et al. 2016 so the methodology was different and the species (zebra finch vs Bengalese where zebras have less complicated syntax). Also see Hara et al. 2007, European Journal of Neuroscience, where unilateral injection into VTA/SN decreased singing rate in female-directed but not undirected song. It should also be noted that manipulation of dopamine input from VTA/SN to Area X can impact song/social context – see papers by Gadagkar Science 2016, et and also LeBlois and Perkel 2010 J Neurosci where dopamine receptor blockers affect vocal variability in adult birdsong.

Given the focus on dopamine signaling in this paper, I recommend a more careful review of the literature about the role of dopamine in Area X and how the authors’ approach provides new information. It can be summarized efficiently if there is a length restriction.

We appreciate the reviewer’s comment and have now expanded our discussion of dopamine and the parallels between the function of dopamine and FoxP2 in the Introduction. Please see lines 53-70 of our submission.

6) Lines 52-57: The topic jumps in this paragraph from Huntington’s disease to FoxP2 manipulation in deafening animals. It would be helpful to have a discussion about the findings of the Murugan 2013 study in which FoxP2 knockdown in Area X has effects on song and social context and dopamine signaling (as per comment #5 above). Then, lead into the novelties of your study and methodology. The power of your study is in the experimental design and execution-the first evidence for FoxP2 rescue in songbirds and a direct role for dopamine signaling/interaction with FoxP2. If it is the first evidence for FoxP2’s role in sequencing, that should also be stated.

We have extensively edited our Introduction and now include a clear description of the Murugan study in the Introduction and a more complete narrative of what has motivated this research.

Results:

1) Line 31: Define the term song linearity as this is its first use (=is it vocal repetition?) vs syntax.

We provide an in-context definition of song linearity (lines 136-140).

2) In addition to measuring repetitions and syntax, the authors examined whether vocal variability was altered by FoxP2 knockdown in Area X by measuring the CV of pitch and entropy. Did they examine phonological measures that might change in song structure such as similarity scores (see point # 2 in Discussion)? How many song renditions and syllables per bird over what length of time were used for the calculations? this information was absent from the Methods.

We have measured acoustic similarity (%similarity score in SAP2011) and sequential similarity (sequential similarity score in SAP2011) to baseline song in FoxP2 knockdown birds and in birds that later had a rescue of FoxP2 knockdown. Please see new Figure S3. Our results confirm large scale changes in song following knockdown and a lack of significant difference after rescue relative to baseline. We have added a subsection

called 'song similarity and variability' in the methods, in which detailed descriptions of these analysis are provided including the number of songs for each condition per bird.

3) Fig. 2b: given the short and similar motifs across the spectrograms, it would be helpful to use letters or numbers underneath the spectrogram to denote the syntax and make it easier to see the switch in syllable order, etc. A Markov chain figure would also help to show the difference in syllable transitions between the control and knockdown groups.

We have used letters to label individual syllables underneath the spectrogram and used lines to indicate motifs and highlighted vocal elements. Difference song matrices are now included in the main Figures 2-3 and Figure S2 to demonstrate the changes in syllable transition probabilities across different conditions relative to baseline. We have also now provided syllable transition probability matrices for baseline, knockdown and rescue timepoints for several birds in Figure S5.

4) Fig. 2d: define motif versus bout vs syllable for the reader in the spectrogram legends so that the naive reader does not have to go back and forth to the Methods text.

We have now included these definitions in the Introduction (lines 36-40).

5) Fig. 2d-e: indicate the time point (one, two or four months) post-virus injection here and whether the other timepoints were not significant between controls and shFoxP2.

We have specified the time point in the legend. We chose to use this time point because most of rescue experiments were carried out after two months. We provided detailed information in Fig S3 where the changes in repetitions at other time points are shown.

6) Fig. 5d-more clarification on the scale below the x-axis is needed.

This plot is no longer included in the manuscript.

7) Fig. 6-explain the left vs right shift of the box plots-is it meaningful or is just the height of the plots meaningful? This information is difficult to parse-can you state more clearly what the top vs bottom trace shows in each panel?

We have added text to the legend for Figure 6 to clarify. Both the shift and height of the plots are meaningful because the plots are showing the distribution of normalized expression within the population. For *Drd1*, more cells in the CS-shScr population have a lower normalized expression, so the trace for this population is shifted leftward with a higher peak at these values than the CS-shFoxP2 population. For *Drd2*, no shift or difference in peaks is seen between the two populations.

8) Line 338-were dopamine infusions done in undirected finches only-clarify.

Also, since you did not find changes in spectral or temporal structure with dopamine infusion/blockers, this is worth comparing to the LeBlois and Perkel 2010, 2012 papers that did find changes in vocal variability particularly with D2 receptors.

We now define undirected singing on line 51 of the manuscript. Our pharmacological manipulations were only undertaken during undirected singing. We now specify this in the Results. Our dopamine infusion experiments were designed to test whether excess dopaminergic signaling in Area X is sufficient to cause phenotypes similar to those observed following FoxP2 knockdown. Thus, we used dopamine and dopamine agonists rather than dopamine antagonists, as used in the Leblois studies. We describe the Leblois results in the Introduction and discuss why we did not undertake directed singing experiments in the Discussion. Given the lack of social

context manipulations in our study and our use of dopamine and dopamine agonists, as opposed to antagonists, it is not possible to make direct comparisons between these different studies.

Discussion:

The Discussion is an excellent and thoughtful summary of the results with respect to how FoxP2-dopamine pathways interact and with relevance to human speech disorders.
Specific comments below-

1) Line 447-Restoring FoxP2 expression later in adulthood resulted in recovery of linear song syntax. This sentence conveys that your rescue phenotype improved syntax yet in the Results and figures, you stated that it only reduced vocal repetitions and that there were still weird syntax changes (including insertion of novel elements).

We have now corrected this misstatement. Thank you for bringing it to our attention. It now reads (line 338):

“Restoring FoxP2 expression later in adulthood resulted in recovery from aberrant vocal repetitions in all cases while alterations in syntax were largely maintained.”

2) Line 482-However, we did not observe spectral degradation of individual song syllables, as seen following deafening.

In order to be confident in this statement, it would be necessary to measure other spectral features of song in knockdown vs control animals besides CV of entropy and pitch presented here. For example, Haesler et al. 2007 and Heston et al. looked at virus-mediated changes in similarity (accuracy) scores in juveniles as well as CV of multiple acoustic features (duration, amplitude, FM). Can you do select further analysis to confirm?

We now provide analysis of pitch CV, entropy CV, %song similarity, %sequential similarity, and the trial-to-trial variability of song similarity and sequential similarity. As the reviewer is likely aware, song degradation following deafening is dramatic and easy to appreciate. Entire syllables start to fall apart and become highly variable and spectrally degraded. We provide several types of analysis and provide ample examples of song spectrograms from birds following Foxp2 knockdown and following rescue. Therefore, we think the combination of raw data and thoroughly analyzed group data in our manuscript fully support our conclusion that “we did not observe spectral degradation of individual song syllables, as seen following deafening”.

3) Line 534: extra ‘to’ in this sentence... birdsong and speech plans into to fluent vocal-motor actions.

Thank you for bringing this to our attention. It has been corrected..

4) It is worth commenting in this discussion on why you did not observe effects of dopamine infusion and receptor blockade on song features given evidence from the LeBlois and Perkel work. What does this mean for the Sasaki et al. 2006 paper which showed that dopamine levels are higher in Area X for FD but not UD?

We did not conduct experiments during directed singing. Please also see our response to #8 of the Results section above.

Methods:

1) Line 698-Typos/extra words in this sentence: scarified at 180-200dph. Birds were put down prior to lights in the to ensure that singing behavior did not

We have corrected these issues and thank you for catching these errors.

Supplements:

1) Fig. S3-clarify the presence of the second black line in panel b and e.

The second black line indicates a second vocal element in this birds' song which also changed its repetition rate following FoxP2 knockdown. We have revised the legend to clarify.

Reviewer #2 (Remarks to the Author):

This paper has 3 main components; (1) effect of FOXP2 knockdown in adult area X on singing behaviour, (2) single nuclei analyses of area x subtypes and consequences of knockdown on dopamine expression and (3) effect of optogenetic stimulation of dopamine circuits on singing behaviour. The individual studies are beautifully executed, however the 3 do not quite come together as a whole. With some revisions and new data analyses it is likely that they could come together more coherently. The main concerns regarding the study are:

1. The single nuclei sequencing study seems far more focused on establishing cell types to say something about the circuitry of area X than looking at the role of FOXP2 or consequences of its disruption. The point of the section on lines 226-286 seems to be trying to prove something about direct- and indirect-like pathways/populations in area X, which is not well supported, nor does it fit with the narrative of the paper. It would be better instead to use this excellent data to draw conclusions about the role of FOXP2 in these cell populations and establish the identity of the populations so that any disruptions to expression patterns could be explored thoroughly after FOXP2 knockdown (see also 2). The paper should also be very careful about drawing parallels with the mammalian direct/indirect pathway system when only relying on sequencing data and without neurophysiological data to support the conclusion that "MSNs in Area X can segregate into broad classes corresponding to direct-like (Drd1+/FoxP2+) and indirect-like pathways (Drd2+/FoxP2-) (Figure 5a), as well as a third Drd1/2+ class.". In fact, just from the sequencing data, I do not see the parallels to direct/indirect categories since that FOXP2 positive cells are prominent in every group, including the DRD2 positive groups in Figure 5.

We agree that caution must be taken when comparing the cell types in Area X to mammalian MSN pathways using only transcriptomic data. Therefore, we have restructured this portion of the Results to focus first on FOXP2 expression within the MSNs, then dopamine receptor expression, and then a potential comparison with mammalian MSN pathways. We have modified the language to clarify that these descriptions are of putative direct-like and indirect-like pathways and added the following clause to the concluding sentence of the paragraph (lines 248-249): "Although a deeper characterization of the physiological properties and connectivity will be required,...". In addition, the title of Figure 5 has been changed to remove reference to direct- and indirect-like pathway organization and now reads "Patterns of FoxP2 and dopamine receptor expression in Area X."

2. The single cell sequencing data is sorely under used to determine consequences of FOXP2 knockdown – only consequences on dopamine receptors are looked at, whereas there is an enormous and valuable dataset that is basically ignored. This data set should be fully exploited to determine the molecular consequences that result from FOXP2 knockdown. Such a fine-grained molecular analysis of FOXP2 disruption has never been reported in birds and would represent an important resource for the field and may give important new insight into the function of FOXP2, rather than limiting the study to only confirm previous findings that FOXP2 regulates dopamine receptors. This is not to malign Figure 6 - it is very well done, but it is unfortunate not to go further with other genes given the dataset at the authors disposal.

We appreciate the comment about the value of the dataset. There are thousands of genes that show differential expression between the knockdown and control datasets, which we make publicly available as supplemental data. We originally focused on dopamine receptor expression to support the rest of the submission, as our goal was to understand how FOXP2 and dopamine may interact in vocal disfluencies. In this revision (lines 333-350), we now lead with a description of differences in FOXP2+ cells between the two experimental groups for major categories of gene expression differences. We highlight genes that have been noted as being most closely linked to autism and show transcription targets of FOXP2 that are altered as a result of the FOXP2 knockdown as well. This new analysis is shown in a new Figure S7.

3. The link between the optogenetic studies and FOXP2 are tenuous. It is not clear that these optogenetic studies tell us anything about FOXP2 circuitry. As it stands the experimental links between the FOXP2 knockdown and consequences of phasic dopamine are somewhat disjointed and coincidental in their relationship, rather than being directly proven. From the text it is not entirely clear to me what can be really taken away from this combination of experiments. If the optogenetic studies were to be combined with the FOXP2 knockdown approaches in the same animal there could be some very exciting proof of function experiments reported.

We have extensively edited the manuscript to more clearly emphasize the links between FoxP2 and dopamine. For example, the Introduction now includes an overview of FoxP2 function and then draws out the parallels with dopamine signaling. Starting on line 34 this part of the Introduction reads:

“FoxP2 is strongly expressed in the songbird Area X, a specialized song nucleus within the striatum that is important for song learning and the control of song variability¹³⁻¹⁷. Its expression is thought to be necessary for learning song during development but not for the maintenance of song in adulthood. Zebra finch song consists of repeated introductory notes followed immediately by ~3-7 syllables that are produced in a highly stereotyped sequence or syntax, referred to as a song motif¹⁸. This song motif is typically repeated two or more times in what is referred to as a song bout. Juvenile male zebra finches learn their song by imitating the song of an adult male bird, typically their father, during development. Knockdown of FoxP2 in Area X of juvenile zebra finches causes a variety of vocal learning deficits, including inaccurate syllable imitation, reduced stereotypy of song syntax, and anomalous repetition of song syllables¹⁰.

FoxP2 appears to have a more restricted role once song is learned. A core feature of zebra finch song is low trial-to-trial variability in the acoustic structure of song syllables¹⁹. Knockdown of FoxP2 expression in Area X of adult birds has not been reported to cause a deterioration of singing behavior or problems with song syntax¹¹. However, FoxP2 expression levels are inversely correlated with trial-to-trial variability of song syllables (low levels of FoxP2 are associated with increased song syllable variability) and FoxP2 has been linked to gene expression modules associated with vocal variability^{12,20-22}. In addition, male zebra finches decrease the trial-to-trial variability of their song by ~1-3% when singing to

a female bird, a performance mode known as directed song, when compared to undirected singing^{18,19}, and knockdown of FoxP2 blocks this social context-dependent decrease in song variability¹¹.

Several lines of evidence indicate that that the function of dopaminergic input to Area X and the behavioral role of FoxP2 expression are tightly linked. Like the mammalian striatum, Area X receives glutamatergic input from pallial/cortical regions and input from dopaminergic neurons in the substantia nigra and ventral tegmental area (VTA)²³. FoxP2 is thought to exert its effects in part by regulating the expression of D1 dopamine receptors in MSNs. Disruptions of FoxP2 expression result in reduced expression of D1 dopamine receptors in Area X¹¹ and have also been shown to increase dopamine levels in the rodent striatum²⁴. Ablating dopaminergic input to Area X severely disrupts song learning in juvenile birds but does not drive deterioration of overall song acoustic structure or syllable syntax in adult birds; these behavioral effects are similar to those associated with disruption of FoxP2 expression in juvenile and adult birds^{10,11,25,26}. Moreover, like disruption of FoxP2 expression, 6-hydroxydopamine lesions of dopamine terminals in Area X reduce trial-to-trial variability of song syllables during undirected singing²⁷, and pharmacological inhibition of D1 receptors blocks social context-dependent decreases in song variability²⁸.

Together, these previous studies suggest that FoxP2, D1 receptors in MSNs, and dopamine input function in concert to regulate song syllable acoustic structure and vocal variability. We were motivated to further explore the function of striatal FoxP2 and dopamine in adult vocalizations because several additional lines of evidence indicate that we still have an incomplete view of their role in controlling vocal motor sequences.”

Our explicit goal in the second half of this study was to test whether manipulations of dopaminergic input to Area X result in behavioral disruptions similar or dissimilar to those that we identified following FoxP2 knockdown. Given the background highlighted above, this question is appropriate and in line with the overall goals of our study. Moreover, given the already numerous novel findings and novel approaches brought to bear in this study, we respectfully disagree that additional and unspecified experiments linking FoxP2 and dopamine signaling are necessary at this juncture. This being said, we agree with the reviewer that combining FoxP2 and dopamine manipulations could be exciting and provide important further insights to the function of these neuromodulatory and gene regulatory pathways in vocal motor control, but we also contend that they are beyond the scope of the current study.

It would be much better to where possible refer to the original research than being overly reliant on review papers.

We have replaced most review papers with the original research where appropriate.

I would like to highlight that the data analyses and figures are really excellently done – the data representation makes the findings very clear and are beautifully represented. Also, the experimental design of the inducible Cre FOXP2 knockdown system is superb.

Thank you for these supportive comments!

Reviewer #3 (Remarks to the Author):

General comments:

This study by Xiao et al on the role of FoxP2 in singing behavior, physiology, and gene regulation in adult songbirds puts lots of pieces of a puzzle together, to come out with exciting findings. With an apparently stronger knock down of FoxP2 in the striatal song nucleus Area X, they found very little acoustic changes, but syntax and stuttering deficits, more like what is seen with humans with a FoxP2 mutation. They are able to recover (repair) the deficits using a Cre reversal vector. They, for the first time, perform single nuclei transcriptome sequencing within the songbird striatum, characterize the cells types more robustly, and make needed advances in relationship with mammals. They find that the knockdown changes the balance in the D1 and D2 receptor families, in manner consistent with the behavioral findings and physiology of the striatum. This leads to a hypothesis of dopaminergic function, which they test with optogenetic stimulation of dopaminergic synthesizing neurons, which causes some similar behavioral deficits as the FoxP2 knockdown. This paper has it all, is multidisciplinary, and makes important advances in our understanding of the role FoxP2 plays in vocal learning circuits, song and speech. I am very excited by the study.

We thank the reviewer for their positive and constructive comments.

I do have some concerns, which I think addressing them will improve the manuscript. The authors were pretty good as relating their findings to the past literature, but missed out on a several relevant important studies.

The authors do not relate their findings to the FoxP2 studies and vocal behavior in rodents. The most relevant are the Castelluci et al 2016 (Scientific Reports) and Chabout et al 2016 (Frontiers) studies. They found that the heterozygous FoxP2 mutant mice with the same or similar mutation as humans had very little to no acoustic changes in their syllables, but had sequencing deficits. The males did not readily switch to the more complex sequences that they make in the presence of females, and instead continued to produce the simpler syllable types, with reduced syntax variability. They suggested that these sequence deficits are similar to the deficits seen in humans, where the underlying problem is difficult for us humans to sequence a complex series of phonemes into complex words. Chabout et al argued that this finding in mice and humans was different than the Heassler et al 2007 study in zebra finches, where the later found mostly acoustic deficits in learning but little to no sequencing deficits. They hypothesized that knocking down FoxP2 in the striatum only (in songbirds) may have a different deficit than heterozygous partial knockdown in the whole brain (humans and mice). Xiao et al now find that they can also get sequencing deficits in adult songbirds with a local striatal knockdown, refuting this explanation for differences between studies. What the combined studies suggest to me is that like Chabout et al stated, FoxP2 may have been utilized for sequential (and possibly acoustic) control of vocalizations by the striatum, before vocal learning evolved; but that when the vocal learning circuitry appeared or became highly specialized in songbirds and humans, the circuit became even more dependent on the gene. I think the authors need to consider these rodent findings and the associated hypotheses in the interpretation of their findings as well.

Thank you for this important thought. We added this to our Introduction (lines 70-73) and in the Discussion in the paragraph starting on line 395-397.

As the authors noted, some of the stuttering findings after adult Area X lesions in the Kubikova et al 2014 (Scientific Reports) is similar to the FoxP2 Area X knockdown stuttering findings in the current study. But, I don't think the authors go far enough in comparing and contrasting these findings. Kubikova et al found stuttering mainly on the motif end syllables, if the bird had a predisposition of some repeats before the lesions, and only with chemical lesions that left fibers intact, and not electrolytic lesions. They further found a relationship between stuttering when new neurons enter Area X to repair the nucleus. They did not find stuttering with a chemical

LMAN lesion; but Chakraborty et al 2017 (Scientific Reports) later found stuttering with viral NR2B knockdown in LMAN, both at the beginning and ends of song motifs. All this is to say that when combined with the findings of Xiao et al, it looks like some manipulations and conditions lead to stuttering, to different types of stuttering, and others do not, including when using different viral vectors to knock down FoxP2. I do believe the author's interpretation that Area X is involved in the initiation and stopping of motifs, and syntax is plausible. But I think multiple things have to line up for this to happen. BTW, the authors should also credit Kubikova et al 2014 for an initial proposal for such an idea "we propose that LArea X may be involved in providing cues for the initiation and selection of the next motor segment in a song sequence in adult animals, and the selection is more critical in individuals with learned repetitions at the end of their motifs".

We have now discussed the Kubikova result, as well as the earlier Area X lesion inducing 'stuttering' in Bengalese finches (Kobayashi et al 2001) in 3 separate places in the manuscript.

In the Introduction (Lines 76-78): *"Third, Area X has been implicated in song initiation, syllable sequencing, and song maintenance. Neurotoxic lesions in Area X can induce transient repetitions of song syllables and problems initiating song bouts (Kobayashi, Uno et al. 2001, Kubikova, Bosikova et al. 2014)."*

In the Results (Lines 130-133): *"CS-shFoxP2+ birds appeared to get caught in 'motor loops', repeating a certain song syllable many times before transitioning to the next syllable. Lesions of Area X can cause birds to transiently increase how many times they repeat introductory notes or other syllables that they already tended to repeat prior to lesions (Kobayashi, Uno et al. 2001, Kubikova, Bosikova et al. 2014)."*

In the Discussion (Lines 409-411): *"When considered within the context of the broader literature, it appears that Area X plays a dual role in song. It is involved in learning how individual song syllables should be sung and in controlling larger scale selection and sequencing of these syllables (Sohrabji, Nordeen et al. 1990, Scharff and Nottebohm 1991, Kobayashi, Uno et al. 2001, Kojima, Kao et al. 2013, Kubikova, Bosikova et al. 2014, Tanaka, Singh Alvarado et al. 2016)."*

However, we do not feel that it is appropriate to cite Kubikova et al for the idea that Area X, a structure embedded in the striatum, is involved in the initiation and selection of motor sequences for song. First, the role of the basal ganglia/striatum in aspects of motor sequence initiation and in sequence selection, and core aspects of this idea for other sequenced behaviors, have been well-established in the mammalian brain. In our view, the songbird research here and in Kubikova et al build from these longstanding ideas and functions to better understand a specific behavior (vocalizations). Second, earlier research by Kobayashi (2001) and Scharff (1991) pre-date the ideas discussed in Kubikova 2014.

It was good to see that the authors measured both D1 and D5 dopamine receptors in the striatal neurons, where many studies only look at D1. I think they should consider the alternative names for these two receptors, D1A and D1B respectively, whose evolution and a revised terminology was mapped out in two other studies Kubikova et al JCN (2010) and Yamamoto et al 2013 (Mole Eco Evol). Both receptors belong to the D1 family, whereas the D2 family has a different evolutionary origin, converging on binding to dopamine. Although the authors did measure that three dopamine receptors that are found in humans (D1, D5, and D2), birds also have three additional receptors (D1C, D3 and D4), with the former two also expressed in the brain, although not the striatum (Kubikova et al 2010). D3 and D4 belong to the D2 family. I think it will be worthwhile mentioning this to the readers.

We appreciate this information and we have added it to the text (paragraph starting on line 229).

How do the authors interpret their findings in the context of those from the White lab (Teramitsu et al 2004 J. Neurosci) that singing causes down regulation of FoxP2 expression in Area X? Considering that they are doing a knockdown.

Although we find the studies from the White lab compelling, showing that singing causes down regulation of FoxP2, we do not think that our current study directly informs this work because we are studying alterations in large scale structure of song occurring and accumulating over many weeks as birds continue to sing. For this reason we would prefer not to make comparisons or speculate about this.

Specific comments:

Line 34. State that Area X is a song nucleus within the songbird striatum.

We have revised as suggested (see line 34).

Figure 1. Panel d: What are the individual values? Where is the triple label image?

Each point in this panel represent the normalized expression level of FoxP2 in BFP+ or mCherry+ cells relative to control cells (FoxP2+mCherry-tagBFP-) from individual slices. We have added this sentence to the legend.

We also provide Figure S1e showing the triple labelling of mCherry, tagBFP and Cre-GFP and 1d showing the triple labelling of mCherry, tagBFP and FoxP2.

Line 193. It is good that the authors describe the variability of findings from individual animals. They should keep such descriptions.

We have tried our best to report such potential outlier from individual experiment within the word limit of the main text. We have now also included syllable transition matrices in Figure S5 to visualize the syntax changes across conditions from individual animals.

Line 215. Here the authors used a paired t-test, and elsewhere use Wilcoxon signed rank or other test when comparing the same animals at two different time points. The authors need to be consistent.

We apologize for this inconsistency. We have used Wilcoxon signed-rank test to replace paired t-test in Figure 3c-d. The conclusions remain the same.

Line 227. I suggest changing from FoxP2 is expressed in "Area X" to "Area X and the surrounding striatum".

Although accurate, we think this is not necessary to convey our main point to the reader.

Line 240. Arkypallidal cells. The author should cite a paper for this cell type (have not heard of it before), and perhaps include a panel that shows the result of the cell type also in Area X.

We have added references and a supplemental figure (S10) showing our evidence.

Line 261. No citation is given to these findings of D1 and D2 overlap in the prior study. Do they mean to cite Kubikova et al 2010?

Although we have edited this section, we do provide citations to the Kubikova 2010 study.

Line 458. But did the prior mammal studies also include D5 (D1B) in their studies? Most have not.

These studies did not include D5 (D1B) and we have added this information (now on line 383).

Line 474. What is the difference between sequences and syntax? I see in the methods, maybe the authors mean changes at the beginning and end of a motif for sequences, and the middle for syntax? This is not a common, nor I think a good, definition. I would call all of this syntax or sequence changes, and their locations defined as end or internal syntax changes.

We now provide definitions in the Methods section pasted below (starting on line 546):

“Quantification of changes in syntax.

Syntax alterations are defined as new syllable transition(s) (i.e., transitions not present prior to FoxP2 knockdown) occurring in a song motif. Syllable omissions are dropped syllables that were present in the song motif prior to FoxP2 knockdown. Syllable additions are as insertion of de novo created syllable(s) (i.e., syllables not present prior to FoxP2 knockdown) into the song motif. Omission and addition of syllables at the beginning or end of a song motif are not classified as syntax alterations. Quantification of percent syllable alteration, omission and addition was calculated from 30 song bouts at each timepoint presented in the figures. Changes in song sequences refers to any change in a song bout which alters either the syntax, syllable transition probability or frequency of vocal repetitions. For example, in the song in Figure 2e syllables ‘ab’ and ‘ghj’ are consistently dropped after FoxP2 knockdown. Syllables ‘ab’ and ‘ghj’ were noted to also be dropped in a subset of motifs at baseline. The consistent dropping of these syllables is considered a change in song sequence. These and other changes in the song are captured by changes in the syllable transition Matrix (Figure 2g) and syllable repetitions but are not considered syntax alterations.”

Line 533. In light of the mouse findings, convergence of FoxP2’s role in vocal sequencing could have been inherited in songbirds and humans, and then enhanced in the vocal learners independently.

We agree that this could be the case, but here we are making a broader point about the role of basal ganglia specializations, beyond those associated with FoxP2, for producing fluent vocalizations.

Line 545. It would be helpful to add a phrase in some of the sentences to indicate what each virus was used for, like FoxP2 overexpression, knockdown, etc. Otherwise the sentences read as if they are one continuous protocol for just one virus construct.

We have added descriptions to distinguish Cre switch constructs for knockdown and pLenti construct for overexpression.

Line 600. Did the authors select good singers before surgeries, and if so, how? I imagine it will be hard to get animals to sing after many of these manipulations.

Birds subjected to optogenetic manipulation were pre-screened because the voice detection algorithm we use cannot do on-line detection of all syllables or syllables from all birds. As described in our previous work (Xiao et al, 2018), the false positive/negative targeting rates need to be maintained under 10%. The birds with songs that

do not meet this threshold were not used. We did not screen birds based on how much they sang or other features, only whether our voice detection algorithm could reliably identify the correct syllable in their song. Otherwise, birds set up for other experiments were randomly assigned. Generally, the singing behavior was not affected after several rounds of surgeries following our protocols.

Line 613. Where the songs for analyses selected manually or automatically?

Song bouts used for analysis were selected randomly from 1-2 days of song at the timepoints specified relative to circuit manipulation.

Line 639. I am not following this paragraph well, but it seems like the authors are filtering out songs without repeats before analyses? That would be a bias?

Based on the birds' singing behavior during our experiments (during FoxP2 knockdown or during optogenetic manipulations), we first identified syllables that were being repeated an unusually high number of times. This behavior is unusual for a zebra finch and simple to identify in spectrograms. We would then systematically track how many times these same syllables were being repeated by the bird at earlier timepoints, like at baseline prior to any manipulation, or forward in time as our manipulations continued or for several months. We are not filtering out any data and agree that this sentence gives an incorrect impression of our data analysis routines. We have removed this sentence.

Line 704. Where the Area X punches made under a microscope? Was it possible to avoid the surrounding striatum? What is the degree of spread of the virus inside and outside of Area X? Did the authors use fluorescent markers to identify the injection site for the punch?

The Area X punches were made under a dissection microscope. We could easily recognize Area X in fresh tissue under the microscope because of its heavy myelination. Given the large size and clearly visible borders of Area X, we are confident that our samples did not contain adjacent portions of the striatum. The expression of our CS constructs was restricted within Area X and it provided reasonable coverage in terms of the spread of AAVs (see Figure S1). We did not use fluorescent markers to identify the injection site for the punches because without further amplification of fluorescent signals using immunostaining, we were concerned that this would lead us to underestimate the borders of Area X and underestimate the extent of our viral labeling.

Line 725. Cite Rhie et al 2020 biorxiv for the new zebra finch assembly used.

We have added this reference.

Figure S1. Looks like the injections respect the boundary of Area X? This would be strange. Area X is also not perfectly round as the dashed lines in panel e indicate. Which direction is front of the brain?

The boundary of Area X appeared as a bright oval when viewed with a DIC objective because of myelination. In most cases, AAVs injections in Area X respect the boundary very well following our protocols. We apologize that the dashed lines we used in panel e is misleading and suggests that Area X is perfectly round. We have revised the shape of dashed lines and indicated the orientation of the brain section in panel e.

Figure S3. I am surprised that the last value at month 5 is not significant in the first graph? Why are errors bars on some months but not others?

The 'N.S.' refers to the repetitions/bout (black axis on the left) at the 5th month compared to the baseline. Error bars are present uniformly while some of them are too small to be seen.

Figure S4. What does the yellow circle mean in panel c? Shouldn't this panel come at the end of the figure?

The yellow circle represents birds that did not display syllable addition or omission either following knockdown (CS-shFoxP2) or rescue (Cre-GFP), thus they are overlapping at the origin. We have revised the legend and moved this panel to the end of this figure.

Figure S7. The green circle in panel c is not the shape of Area X.

We have revised the figure to better indicate the border of Area X and IMAN.

Reviewers' Comments:

Reviewer #1:

Remarks to the Author:

The authors did an excellent job revising the Introduction to provide the prior history leading up to their work and how their work expands upon our knowledge of interactions between FoxP2, Area X, and dopamine using a multi-faceted approach. Their Introduction and Discussion have also been strengthened by their inclusion of human and mouse data with relevance to their findings for apraxia of speech which will attract readers in that area. The flow of the revised manuscript text and figures is now excellent. The authors should be commended for their revision efforts and attention to detail. No further revisions are necessary.

Note, a few minor grammatical mistakes were found as follows in the main manuscript-

Line 53: extra 'that' inserted: Several lines of evidence indicate that that the function of dopaminergic input to Area X and the

Line 67: spelling of 'motivated': in concert to regulate song syllable acoustic structure and vocal variability. We were motivated

Line 143 extra word 'Restoring' in this sentence: In addition to disrupting song Restoring, we found syntax changes in 50% of CS-shFoxP2+ birds (4 of 8

Line 236 extra word 'of' also showed distinct patterns of within the MSNs: 36% of the 9,672 MSNs were exclusively Drd1+ and/or

Reviewer #2:

Remarks to the Author:

Xiao et al have thoroughly addressed the comments of the reviewers and strengthened the manuscript. In particular the expanded introduction now makes the logic of addressing both FOXP2 knockdown and dopaminergic manipulations much clearer. The more in-depth analyses of the single cell data are also highly appreciated. The manuscript now seems appropriate for Nature Communications, with a few minor issues to be addressed before publication, as listed below in order of appearance. All line numbers refer to the numbers in the PDF version.

- On line 30: Reference #6 is cited, but this seems to be an error as no such paper exists in AJHG. Replace with MacDermott et al. Identification of FOXP2 truncation as a novel cause of developmental speech and language deficits. AJHG. 2005
- Provide a reference for the sentence on lines 35-36
- Provide references for the sentence on lines 41-43. Only Haesler et al 2007 is mentioned. Also cite Norton et al, J Neurosci, 2019, Adam et al, Mol Cell Neuro, 2016 and Schulz et al, Genes Brain Behav 2010
- Please ensure that all datasets are made publicly available in full, particularly the single cell dataset.
- Please make very clear and explicit in the text where the Drd1/2/5 expression is expressed what nomenclature is followed. Another reviewer raised this point and can be confusing which receptors are equivalent between birds and mammals, so it would be very helpful to have this clearly explained in the results where the data is discussed.
- I strongly recommend a summary graphic be added to the end of the paper. A lot of complex findings are presented in this paper and it would be extremely beneficial for the reader to have a summary of what was found using the different approaches, how the behavioural effects compare when FOXP2 or dopamine signalling is manipulated, and the proposed model of action.
- Lines 338-340 seems to only reference dopamine or basal ganglia studies. This should also reference Foxp2 studies I believe.
- Lines 351-2: reference the studies on FOXP2 mutations in CAS.

- Lines 419-20: Mention and reference van Rhijn et al, Brain struct func, 2018 with regard to Foxp2 mutations causing inhibitory/excitatory imbalances in striatum in mutant mouse models.
- Lines 436-39: This concluding statement comes a little out of nowhere bringing in volitional control as a terminology and fundamental disruption. Volitional control of vocalisations means different things in different communities/species and can be quite unclear what is being referred to. In addition, this terminology is never mentioned earlier in text with regard to the findings. The text should either define and discuss volitional control earlier in the manuscript, explaining clearly what you mean by the term and how your data supports this statement, or leave it out.

Reviewer #3:

Remarks to the Author:

The authors were very responsive to my recommendations. The paper is much improved, and represents a milestone in understanding the role of FoxP2 in learned vocal behavior. They even added some changes that they said they would not add in the response letter, but still did so in the main text. The revisions and additional data are sufficiently thorough, that for one of the rare times, I do not have any more comments of concern. I think the study is about ready for publication.

REVIEWERS' COMMENTS

Reviewer #1 (Remarks to the Author):

The authors did an excellent job revising the Introduction to provide the prior history leading up to their work and how their work expands upon our knowledge of interactions between FoxP2, Area X, and dopamine using a multi-faceted approach. Their Introduction and Discussion have also been strengthened by their inclusion of human and mouse data with relevance to their findings for apraxia of speech which will attract readers in that area. The flow of the revised manuscript text and figures is now excellent. The authors should be commended for their revision efforts and attention to detail.

No further revisions are necessary.

Note, a few minor grammatical mistakes were found as follows in the main manuscript-

Line 53: extra 'that' inserted: Several lines of evidence indicate that that the function of dopaminergic input to Area X and the

Extra 'that' is removed.

Line 67: spelling of 'motivated': in concert to regulate song syllable acoustic structure and vocal variability. We were motivated

Typo is corrected.

Line 143 extra word 'Restoring' in this sentence: In addition to disrupting song Restoring, we found syntax changes in 50% of CS-shFoxP2+ birds (4 of 8

"Restoring' is replaced by 'linearity'.

Line 236 extra word 'of' also showed distinct patterns of within the MSNs: 36% of the 9,672 MSNs were exclusively Drd1+ and/or

Extra 'of' is removed.

Reviewer #2 (Remarks to the Author):

Xiao et al have thoroughly addressed the comments of the reviewers and strengthened the manuscript. In particular the expanded introduction now makes the logic of addressing both FOXP2 knockdown and dopaminergic manipulations much clearer. The more in-depth analyses of the single cell data are also highly appreciated. The manuscript now seems appropriate for Nature Communications, with a few minor issues to be addressed before publication, as listed below in order of appearance. All line numbers refer to the numbers in the PDF version.

- On line 30: Reference #6 is cited, but this seems to be an error as no such paper exists in AJHG.

Replace with MacDermott et al. Identification of FOXP2 truncation as a novel cause of developmental speech and language deficits. AJHG. 2005

Reference #6 is corrected as suggested.

- Provide a reference for the sentence on lines 35-36

References are provided.

- Provide references for the sentence on lines 41-43. Only Haesler et al 2007 is mentioned. Also cite Norton et al, J Neurosci, 2019, Adam et al, Mol Cell Neuro, 2016 and Schulz et al, Genes Brain Behav 2010

Norton et al, J Neurosci, 2019 is added as a citation, we believe the other two papers mentioned by the reviewer are not relevant to the point made in this sentence and have not cited those papers.

- Please ensure that all datasets are made publicly available in full, particularly the single cell dataset.

All datasets are publicly available. This information is included in the Data Availability section

- Please make very clear and explicit in the text where the Drd1/2/5 expression is expressed what nomenclature is followed. Another reviewer raised this point and can be confusing which receptors are equivalent between birds and mammals, so it would be very helpful to have this clearly explained in the results where the data is discussed.

We have added a statement in the results section clarifying the nomenclature.

- I strongly recommend a summary graphic be added to the end of the paper. A lot of complex findings are presented in this paper and it would be extremely beneficial for the reader to have a summary of what was found using the different approaches, how the behavioural effects compare when FOXP2 or dopamine signalling is manipulated, and the proposed model of action.

Summary figure is added as Fig.9.

- Lines 338-340 seems to only reference dopamine or basal ganglia studies. This should also reference Foxp2 studies I believe.

References of FoxP2 studies are added.

- Lines 351-2: reference the studies on FOXP2 mutations in CAS.

Reference is added.

- Lines 419-20: Mention and reference van Rhijn et al, Brain struct func, 2018 with regard to Foxp2 mutations causing inhibitory/excitatory imbalances in striatum in mutant mouse models.

van Rhijn et al.'s work is mentioned and referenced.

'However, it remains to be determined whether identified behavioral abnormalities are direct consequences of imbalances in inhibitory/excitatory circuits or direct/indirect-like pathways.'

- Lines 436-39: This concluding statement comes a little out of nowhere bringing in volitional control as a terminology and fundamental disruption. Volitional control of vocalisations means different things in different communities/species and can be quite unclear what is being referred to. In addition, this terminology is never mentioned earlier in text with regard to the findings. The text should either define and discuss volitional control earlier in the manuscript, explaining clearly what you mean by the term and how your data supports this statement, or leave it out.'

We agree with the reviewer and have changed this concluding statement to read:

"Together, our findings in Area X indicate that vocal dysfluencies in songbirds may be associated with hyperdopaminergic signaling in the striatum, as seen in human vocal dysfluencies."

Reviewer #3 (Remarks to the Author):

The authors were very responsive to my recommendations. The paper is much improved, and represents a milestone in understanding the role of FoxP2 in learned vocal behavior. They even added some changes that they said they would not add in the response letter, but still did so in the main text. The revisions and additional data are sufficiently thorough, that for one of the rare times, I do not have any more comments of concern. I think the study is about ready for publication.